# Single-cell RNA-sequencing reveals early mitochondrial dysfunction unique to motor neurons shared across *FUS*- and *TARDBP*-ALS

Christoph Schweingruber [1,2], Jik Nijssen [2,3], Jonas Mechtersheimer [4], Stefan Reber [4], Mélanie Lebœuf [1,2], Niamh L. O'Brien [4], Irene Mei [1], Erin Hedges [4], Michaela Keuper [5], Julio Aguila Benitez[3], Vlad Radoi [1], Martin Jastroch [5], Marc-David Ruepp [4,6] ✉ & Eva Hedlund [1,2,3,6] ✉

Mutations in FUS and TARDBP cause amyotrophic lateral sclerosis (ALS), but the precise mechanisms of selective motor neuron degeneration remain unresolved. To address if pathomechanisms are shared across mutations and related to either gain- or loss-of-function, we performed single-cell RNA sequencing across isogenic induced pluripotent stem cell-derived neuron types, harbouring FUS P525L, FUS R495X, TARDBP M337V mutations or FUS knockout. Transcriptional changes were far more pronounced in motor neurons than interneurons. About 20% of uniquely dysregulated motor neuron transcripts were shared across FUS mutations, half from gain-of-function. Most indicated mitochondrial impairments, with attenuated pathways shared with mutant TARDBP M337V as well as C9orf72-ALS patient motor neurons. Mitochondrial motility was impaired in ALS motor axons, even with nuclear localized FUS mutants, demonstrating shared toxic gain-of-function mechanisms across FUS- and TARDBP-ALS, uncoupled from protein mislocalization. These early mitochondrial dysfunctions unique to motor neurons may affect survival and represent therapeutic targets in ALS.

Amyotrophic lateral sclerosis (ALS) is a devastating disease with no cure or effective treatments which is defined by the selective loss of corticospinal motor neurons and somatic motor neurons in the brainstem and spinal cord, that innervate skeletal muscles, with resulting spasticity, muscle atrophy and paralysis[1–3]. The majority of ALS cases appear sporadic with unknown etiology, but the disease can also be inherited and caused by mutations in several genes such as superoxide dismutase 1 (SOD1), the DNA/RNA-binding proteins TDP-43 (encoded by the *TARDBP* gene) or FUS and C9orf72[4–8]. Irrespective of the disease causation, selective degeneration of somatic motor

neurons is seen, which has been attributed to their high firing rate, long axons, extensive branching with up to 2000 muscle fibers, high energy demand and vulnerability to ER stress[1,9,10]. Mutations in *FUS* cause about 1-5% of familial ALS cases and are present in a number of sporadic ALS patients[5,6,11]. FUS is a nucleocytoplasmic shuttling DNA- and RNA-binding heterogeneous ribonucleoprotein (hnRNP) which partakes in a broad range of cellular functions. Most FUS-linked ALS mutations destroy the protein's nuclear localization signal (NLS), resulting in cytoplasmic mislocalization, accumulation and formation of large cytoplasmic inclusions that sequester many other proteins and

[1]Department of Biochemistry and Biophysics, Stockholm University, Svante Arrhenius v. 16C, 106 91 Stockholm, Sweden. [2]Department of Cell and Molecular Biology, Karolinska Institutet, Biomedicum, Solna v. 9, 171 77 Stockholm, Sweden. [3]Department of Neuroscience, Karolinska Institutet, Biomedicum, Solna v. 9, 171 77 Stockholm, Sweden. [4]UK Dementia Research Institute Centre at King's College London, Institute of Psychiatry, Psychology and Neuroscience, King's College London, Maurice Wohl Clinical Neuroscience Institute, 5 Cutcombe Rd, SE5 9RX London, United Kingdom. [5]Department of Molecular Biosciences, The Wenner-Gren Institute, Stockholm University, Svante Arrhenius v. 20C, 106 91 Stockholm, Sweden. [6]These authors contributed equally: Marc-David Ruepp, Eva Hedlund. ✉e-mail: marc-david.ruepp@kcl.ac.uk; eva.hedlund@dbb.su.se

RNA species in neurons and glial cells[5,6,12]. Several studies point to toxic gain-of-function (GOF) of FUS in the cytoplasm. Cytoplasmic expression of FUS in mice causes motor neuron degeneration while FUS knockout or wild-type FUS expressed at physiological level, and localized mainly to the nucleus, leave motor neurons unharmed[13–15]. FUS in the cytoplasm may drive phase transition and solidify FUS-containing RNA granules over time[14,16] or it may augment the cytosolic functions of FUS in mRNA stabilization, transport and local translation[17]. Mutations in *TARDBP*, cause ALS in 3% of familial and 2% of sporadic cases, in which TDP-43 is mis-localized from the nucleus to the cytoplasm of neurons and glial cells[8,18–21]. Misexpression or overexpression of wild-type or ALS-mutant *TARDBP* in mice results in neuronal toxicity and motor neurons loss in the absence of cytoplasmic inclusions, which indicates that cytoplasmic aggregation is not a prerequisite for a neurodegenerative phenotype[22,23]. Depletion of TDP-43 from cells causes massive effects on RNA metabolism, as it maintains faithful gene expression by repressing the inclusion of non-conserved cryptic exons[24], e.g., in *UNC13A* and *STMN2*[25–27]. TDP-43 mislocalization to the cytoplasm is seen in the majority (> 95%) of ALS patients[28–30], and may indicate nuclear loss-of-function of TDP-43, expanding the relevance of RNA metabolism to most ALS cases. It is currently unclear how mutations in *FUS* and *TARDBP* lead to selective motor neuron death. We therefore aimed to unravel if there are any shared pathways that link these different genetic etiologies of ALS and if these uniquely affect motor neurons. For this purpose, we used induced pluripotent stem cells (iPSCs), which can model disease phenotypes in ALS and intracellular dysfunctions, including protein aggregation and axonal transport defects[31–33]. Cross-comparison of patient-derived iPSC lines can be complicated by differences in genetic backgrounds, which may obscure subtle changes when several lines are compared across mutations. We therefore used isogenic CRISPR/Cas9 edited lines where each ALS-causative mutation was introduced onto the same control iPSC line. Here we report the single cell RNA sequencing across neuron types derived from isogenic lines harboring FUS P525L or R495X point mutations, knockout (KO) of *FUS* or the *TARDBP* M337V mutation. Collectively our findings reveal early changes in molecular networks associated with mitochondrial dysfunction in motor neurons but not in interneurons across *FUS* mutant lines that are related to a GOF and shared with the mutant *TARDBP* line, in addition to being present in iPSC lines from individuals with *C9orf72*-linked ALS. Coherent with the transcriptomic changes, we observed a decrease in mitochondrial motility across ALS motor axons caused by GOF independent of protein mislocalization. Taken together, our data support an exceptionally early convergence of motor neuron intrinsic dysfunction and provide the molecular basis for therapeutic approaches improving mitochondrial activity in ALS.

## Results

### Isogenic mutant ALS-FUS lines reveal early cell-type specific transcriptional dysregulation

To understand how *FUS*-ALS affects human motor neurons and interneurons differentially and discern GOF versus loss-of-function (LOF) mechanisms, we aimed to introduce aggressive mutations in human iPSCs and compare these lines to a *FUS* KO. To select *FUS* mutations for this purpose, we aggregated clinical features from 502 published ALS cases with *FUS* mutations to conduct a meta-analysis (Supplementary Fig. 1a, and Supplementary Data 1). The results affirmed that the FUS P525L mutation results in the earliest onset (at age of 21 ± 8 years, mean ± sd) followed by FUS R495X (at age of 31 ± 11 years) and both mutations progress rapidly to respiratory failure or death within 17 ± 10 and 20 ± 18 months, respectively (Supplementary Fig. 1b, c) and were selected for our studies. In contrast, most other *FUS* mutations lead to disease in midlife or later (age of 41 ± 18 years). CRISPR/Cas9-mediated genome editing was used to introduce a homozygous *FUS* R495X mutation into the DF6-9-9T.B control line (Supplementary

Fig. 2a, b) and compared to a *FUS P525L* mutation (heterozygous or homozygous)[12] which were generated on the same background. Introduction of the mutations was confirmed by dye terminator sequencing following PCR amplification of the edited loci and assessed the top five off-target loci (Supplementary Data 2). We also used a *FUS* KO line which had been generated on the same background that employs a gene trap in the first intron of *FUS*[34] (Fig. 1a). The pluripotency of the generated iPSC lines was established by immunofluorescence microscopy using antibodies against POU5F1 (OCT-3/4) and NANOG (Supplementary Fig. 2c). Immunofluorescent staining against FUS in iPSCs confirmed the loss of FUS in the KO line, and the redistribution of FUS from nucleus to cytoplasm in the R495X and P525L lines. The homozygous lines showed a complete redistribution of FUS from nucleus to cytoplasm, while the heterozygous P525L line showed expression in both nucleus and cytoplasm, as expected (Supplementary Fig. 2d). To dissect the role of FUS' LOF and GOF in lower motor neuron vulnerability, isogenic iPSC lines were specified to spinal motor neurons and interneurons using two distinct protocols. The first protocol generates 30–40% of interneurons[35], here termed the heterogeneous protocol (Supplementary Fig. 3a, f), while the second protocol generates 60–80% of motor neurons[32,36,37], here termed the homogeneous protocol (Supplementary Fig. 3c, g). Differentiation of iPSC lines showed that motor neurons (ISL1/2⁺) and interneurons were readily generated (Supplementary Fig. 3b, d) and that the proportions of specific neuronal populations and progenitors were consistent across lines, with no marked effect of the mutations on cell type specification (Supplementary Fig. 3f, g)[38–40]. Thus, we have generated isogenic iPSC lines in which FUS LOF and GOF in ALS can be studied across mutations and neuron types to reveal why motor neurons are particularly vulnerable and if there are common pathways that are dysregulated across mutations.

Following characterization of the isogenic FUS mutant and KO lines, we subjected differentiated cultures to FACS to isolate individual live cells and conducted Smart-seq2 single cell RNA sequencing (Fig. 1a). After quality control (see "Methods" section), 1415 cells remained, which were classified according to a decision tree into motor neurons, interneurons and neuronal progenitor cells (Supplementary Fig. 3e). Cells were plotted using UMAP (uniform manifold approximation and projection), which demonstrated clustering of motor neurons, interneurons and neuronal progenitors (Fig. 1b). Cells originating from the different isogenic lines distributed across all clusters (Fig. 1c) consistent with the observation that all lines generate neurons at similar proportions (Supplementary Fig. 3f, g). UMAP projection of neurons only, after removal of progenitors, showed a clear separation of motor neurons and interneurons with clustering of V2a interneurons across lines (Fig. 1d, e). V2a excitatory interneurons are also prone to degeneration in ALS as shown in SOD1^G93A mice[41] and appear more vulnerable than V1 inhibitory interneurons to early disease processes[42]. Thus, we separated these from other interneuron types in cross-comparisons with motor neurons to delineate potential gradients in vulnerability. General neuronal markers (*MAPT*, *NEFM*, *SNAP25*) were highly expressed across neuron types while markers of proliferating cells (*SOX2*, *MKI67*, *CDK1*) were absent (Fig. 1f). Motor neurons showed expression of *ISL1*, *ISL2*, *MNX1*, *UTS2*, *SLC18A3* (*VACHT*), and *SLC5A7* (Fig. 1f, and Supplementary Fig. 3h), while the presence of glutamatergic markers combined with *SOX14* and *VSX2* (*CHX10*) demarked V2a interneurons (Fig. 1g). To further solidify the expression of *CHAT* in the motor neurons we conducted deeper RNA sequencing of bulk motor neurons generated in the homogenous protocol. Then we analysed the reads covering the nested CHAT/ SLC18A3 locus, which confirmed that both genes are expressed (Supplementary Fig. 3i). Immunofluorescent staining against CHAT protein showed that ISL⁺ cells express CHAT protein (Supplementary Fig. 3j). To further delineate interneuron populations generated selected neuronal markers genes were used for hierarchical clustering which

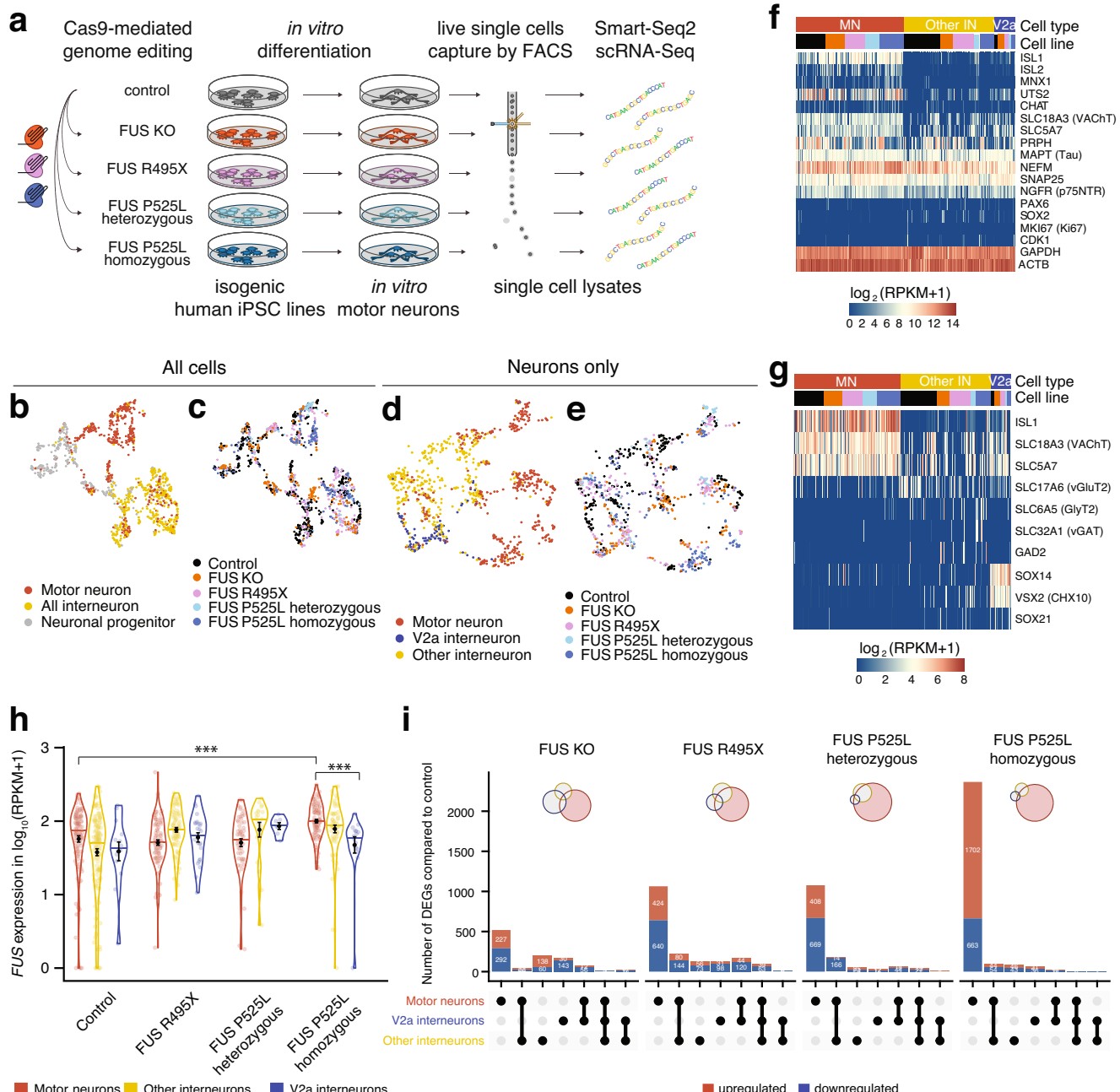

**Fig. 1 | Single-cell RNA-Sequencing identifies RNA dysregulation specific to motor neurons with *FUS* mutations in a dose-dependent manner. a** Schematic showing generation of the different iPSC-lines using CRISPR-Cas9 gene editing, subsequent differentiation into motor neurons and interneurons, sorting and single-cell RNA sequencing. **b**–**e** UMAP projections of all cells of the included cell lines that passed quality control. The two UMAPs on the left are colored by (**b**) assigned cell type or (**c**) cell line, the two on the right contain neuronal cells only, again colored by assigned (**d**) cell type or (**e**) cell line. **f** Gene expression heatmaps of all neuronal cells, showing expression data of several (motor-) neuronal marker genes, progenitor markers and housekeeping genes. **g** Gene expression heatmaps of specific motor- and interneuron subtype marker genes. **h** Violin plots of the *FUS* mRNA expression in the control and ALS mutant lines across cell types. The

normalized expression is given as $\log_{10}$(RPKM + 1). Differential expression was performed within cell type, across lines. Error bars show mean ± standard error of the mean (SEM). * $p < 0.05$, ** $p < 0.01$, *** $p < 0.001$, derived from two-sided DEA testing using DESeq2 and corrected for an FDR of 0.05. The *p*-values are 6.48E-5 for control vs. FUS P525L in motor neurons, and 1.86E-4 for FUS P525L homozygous motor neuron vs. FUS P525L homozygous V2a. **i** Upset plots of the intersections between DEGs in motor neurons, V2a interneurons and other interneurons between the *FUS* mutant lines versus the control cell line. Each column represents a section of a Venn diagram of the cell types as indicated in the table below. The corresponding Venn diagram is shown in the insets. The red (top) and blue (bottom) columns represent upregulated and downregulated genes, respectively. Genes with an adjusted *p*-value of <0.05 were considered differentially expressed.

identified several subclusters, including *VSX2*⁺ V2a interneurons, *ASCL1*⁺*VSX2*⁻ interneurons (Supplementary Fig. 4a). Neurotransmitter analysis also demonstrate clustering of *GAD1*, *GAD2*⁺ subpopulations, containing GABA transporters. Subpopulations expressing glycine and/or glutamate transporters and/or tyrosine hydroxylase,

demonstrate the presence of both inhibitory and excitatory interneuron populations (Supplementary Fig. 4b). The expression of GAD1 was confirmed by immunofluorescent staining demonstrating GAD1 protein in both interneurons (ISL1/2⁻ cells) and motor neurons (ISL1/2⁺ cells) (Supplementary Fig. 4c). In conclusion, we find that single cell

RNA sequencing can readily identify bona fide motor neurons and interneuron subpopulations derived from the isogenic iPSC lines and that neither *FUS*-ALS mutations (heterozygous or homozygous) nor FUS KO impair the generation of these cell types. Thus, we proceeded to investigate if the *FUS* mutations and FUS KO affect neurons in a cell-type specific manner. First, we established that *FUS* mRNA remained at comparable levels with only small changes across cell types in the different cell lines. *FUS* mRNA was only slightly upregulated in FUS P525L homozygous motor neurons compared to control (Fig.1h, Supplementary Data 3). We then performed differential gene expression analysis (DEA) in motor neurons, V2a interneurons and other interneurons between FUS mutations and the control line in our single cell data. In each cell line, we found that motor neurons displayed unique transcriptional changes associated with the individual mutation and little overlap with V2a and other interneurons, suggesting that most of the transcriptional adaptation and dysregulation events are specific to motor neurons (Fig. 1i). Furthermore, motor neurons demonstrated much larger transcriptional responses to the *FUS* mutations and loss of FUS than the interneuron groups. V2a interneurons showed a surprisingly small response to the FUS mutations compared to the other, presumably more resilient, interneuron group. This analysis demonstrated that introducing the R495X or the P525L mutations resulted in a stronger transcriptional response than knocking out FUS. This is clearly indicative of both LOF and GOF being altered in mutant FUS lines. Furthermore, homozygous introduction of the FUS P525L mutation resulted in 2365 differentially expressed genes (DEGs) unique to motor neurons, while the same heterozygous mutation induced 1075 DEGs, showing a dose-dependent response to this gene mutation (Fig.1i, Supplementary Data 4). The homozygous FUS R495X mutation induced 1064 DEGs, which is less than half of that elicited by the more aggressive FUS P525L mutation, indicating that the transcriptional dysregulation is correlated to the severity of the mutation (Fig. 1i). In conclusion, our analysis demonstrates that each neuron type exhibits unique transcriptional responses to *FUS* mutations and the FUS KO. It also shows that *FUS* mutations elicit the largest transcriptional response in motor neurons and that this appears correlated with the aggressiveness of the mutation, gene dosage, as well as FUS mislocalization, and may underlie the specific vulnerability of these neurons to *FUS*-ALS.

### Single cell RNA sequencing points to shared gain-of-function pathways in motor neurons across FUS mutations

Most ALS-associated *FUS* mutations result in a change in subcellular distribution and cytoplasmic accumulation of the mutant protein, which is thought to be toxic[38]. Therefore, we first validated mutant FUS mislocalization in motor neurons derived from the genome edited lines by immunofluorescence, which showed cytoplasmic accumulation of FUS P525L in ISL1/2[+] neurons, unlike for the control line. The P525L motor neurons also retained some signal in the nucleus, similar to what has been seen for other *FUS* mutations that interfere with the NLS[39,40]. We also confirmed the ablation of FUS in the FUS KO motor neurons (Fig. 2a, Supplementary Fig. 5a). We used antibodies with specific epitopes spanning the FUS P525 position (wild-type P525P and mutant P525L) to delineate heteroallelic mutant and wild-type FUS in immunofluorescence staining (Supplementary Fig. 5b, d). The specificity of the antibodies was confirmed by western blots of whole cell lysates from iPSCs (Supplementary Fig. 5c). We found that in FUS P525L heterozygous motor neurons, wild-type FUS localized mainly to the nucleus, while mutant FUS P525L localized to both cytoplasm and nucleus. In homozygous FUS P525L and in FUS R495X motor neurons, we found a clear mislocalization of mutant FUS. To quantify the degree of FUS mislocalization (Fig. 2a), we performed quantitative high-throughput immunofluorescence microscopy. Here, we first delineated the intracellular compartments by nuclear Hoechst33342 and cytoplasmic NEFM staining in motor neurons marked by ISL1[+] nuclei,

and then measured intracellular FUS levels (Fig. 2b) and the degree of FUS mislocalization within mutant ALS motor neurons in 3D image stacks (Fig. 2c). Overall cellular FUS levels remained stable across mutant motor neurons except for FUS KO motor neurons in which FUS levels diminished significantly (Fig. 2b). The nucleocytoplasmic ratio remained intact in FUS R244C motor neurons, but decreased significantly in FUS R495X and FUS P525L motor neurons. FUS P525L heterozygous motor neurons retained more nuclear FUS than those from the homozygous P525L line (Fig. 2c), indicative of a gene dosage effect. The similar level of protein mislocalization in the R495X and the P525L homozygous lines indicates that while the number of DEGs identified across lines (Fig. 1i) correlates with the severity of the mutations in patients (Supplementary Fig. 1) it does not correlate with the degree of mislocalization.

However, the impact on gene expression does correlate with the severity of the mutations in patients (Supplementary Fig. 1) and could be a consequence of differences in their intrinsic biochemical properties such as RNA-binding[43], phase transitions[44] and oligomerisation[45].

To clarify whether motor neuron-specific DEGs were unique to the individual *FUS* mutations or shared across lines, we interrogated the transcriptomes across the lines. We found 471 DEGs that were commonly altered among FUS R495X and the P525L heterozygous as well as homozygous lines, indicative of a shared transcriptional response. Of those, 116 DEGs were further shared with the FUS KO motor neurons and are thus associated with a LOF in *FUS*-ALS (Fig. 2d, red). Thus, 355 DEG were unique to mutant FUS motor neurons and can therefore be classified as bona fide *FUS*-ALS associated GOF changes (Fig. 2d, yellow). Pathways associated with these changes were assessed by gene set enrichment analyses (GSEA), leading to the identification of a total of 655 biological processes changed in response to the investigated FUS mutations. Of these, 81 were shared across mutant *FUS*-ALS motor neurons, and eleven were associated with a *FUS*-ALS LOF (Fig. 2e, red) while 70 represent *FUS*-ALS GOF (Fig. 2e, yellow). We found only nine biological processes that were unique to the FUS KO line, reflecting that the majority of LOF changes occur also in ALS lines, but some may be uniquely related to the greater loss of FUS in the knockout line (Fig. 2f). The processes linked to LOF in *FUS*-ALS motor neurons included RNA splicing, establishment of RNA localization and nuclear export together with processes related to synapse organisation and activity, dendritic spine development and neuronal projections (Fig. 2g). Altogether, this comparative analysis suggests that most *FUS*-ALS dysregulated processes are due to FUS GOF and include metabolic pathways e.g., downregulation of aerobic respiration, oxidative phosphorylation, and mitochondrial respiratory chain complex assembly, regulation of RNA metabolism e.g., upregulation of mRNA 3′ end processing, regulation of RNA splicing, regulation of mRNA processing, RNA localization, and RNA export from nucleus. Furthermore, upregulation of processes involved in organelle and vesicle turn-over e.g., endoplasmatic reticulum tubular network organization, establishment of organelle organization, and Golgi vesicle transport among others were also identified (Fig. 2h). We also employed commercially curated gene ontology sets to independently identify dysregulated pathways in our data. Using Qiagen's Ingenuity pathway analysis[46] with DEGs between mutant FUS and control motor neurons, we identified 59 pathways that were significantly altered in *FUS*-ALS motor neurons at a false discovery rate (FDR) cut-off of < 0.1, and not at all in V2a interneurons and to a very low extent in the other interneurons (Fig. 2i, Supplementary Data 5). Among the top ten pathways identified in this analysis were mitochondrial dysfunction and oxidative phosphorylation, synaptogenesis signalling pathway, as well as pathways regulating translation and protein stability (EIF2 signalling, regulation of eIF4 and p70S6K signalling, mTOR signalling and protein ubiquitination pathway). In conclusion, our single cell analysis points to shared pathways that may underlie very early *FUS*-ALS pathological mechanisms, most

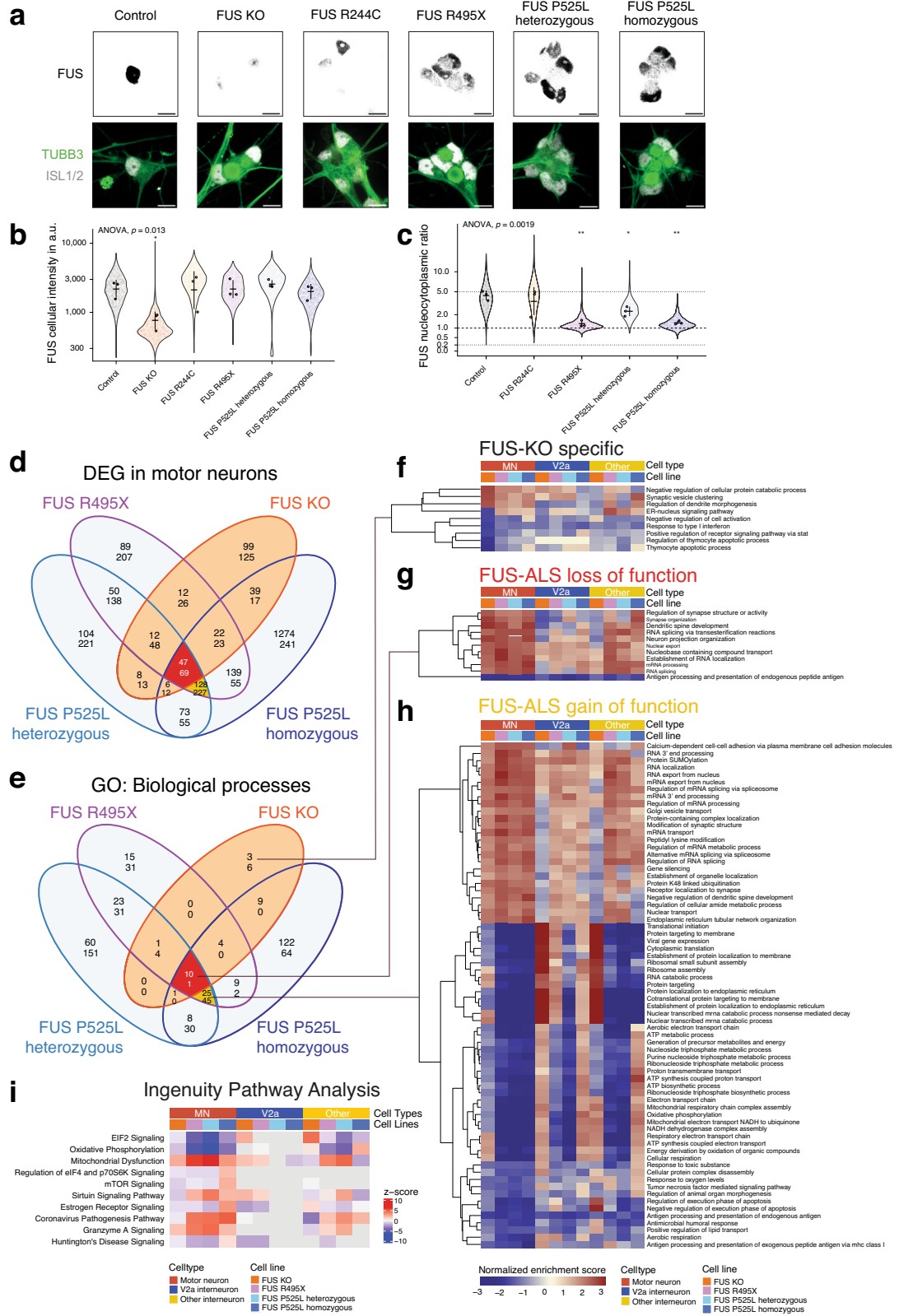

of which are associated with FUS GOF, and which in part appear uncoupled with the level of protein mislocalization.

**The ALS-causative TARDBP M337V mutation leads to massive transcriptional dysregulation unique to motor neurons**

Similar to FUS, TDP-43 is an RNA/DNA-binding protein and ALS-causative mutations in the *TARDBP* gene give rise to broad effects on RNA metabolism[24,27,47,48]. Furthermore, as TDP-43 inclusions represent a pathological hallmark of the majority ( > 95%) of ALS cases[28–30], RNA dysregulation is implicated in ALS irrespective of causation[49]. We reasoned that an analysis of *TARDBP*-ALS may give clues to downstream disease mechanisms across ALS cases more broadly. We thus generated another isogenic iPSC line (on the DF6-9-9T.B control background) that harbored the ALS-causing TARDBP M337V mutation

**Fig. 2 | Shared dysregulated pathways in motor neurons across FUS mutations link mainly to gain-of-function. a** Confocal fluorescence microscopy with immunostaining of N-terminal FUS in motor neurons (TUBB3⁺, ISL1/2⁺) at day 28 reveals cytoplasmic accumulation of FUS R495X as well as P525L and ablation in the FUS KO line. Schematics with representative patterns of different nuclear and cytoplasmic FUS protein localization in the cell lines used in this study. Scale bars are 10 μm. **b, c** The intracellular FUS localization was quantified by high-throughput immunofluorescence microscopy by measuring the total cellular (**b**) and compartmentalized FUS levels in ISL1⁺ motor neurons in 3D image stacks, followed by calculation of the nucleocytoplasmic ratio of FUS levels (**c**). Each dot in the violin plot represents measurement in a single cell, the outlined points are the average levels in three independent differentiation rounds using the homogenous protocol. Bar and whiskers show the mean ± standard deviation across the independent differentiation rounds. ANOVA was conducted on average levels in the three independent differentiation rounds across cell lines and post-hoc pairwise two-sided t-tests between the control and mutant lines. * $p < 0.05$, ** $p < 0.01$, *** $p < 0.001$. In panel b, the p-values are 0.012 (FUS KO), 0.917 (FUS R244C), 0.965 (FUS R495X), 0.4557 (FUS P525L heterozygous), and 0.699 (FUS P525L homozygous). In panel C, the p-values are 0.5607 (FUS R244C), 0.0018 (FUS R495X), 0.0147 (FUS P525L heterozygous), and 0.0053 (FUS P525L homozygous). **d** Venn diagram shows the overlap between DEGs in the different FUS mutant and control motor neurons. Genes with an adjusted p-value of < 0.05 (two-sided DEA test with DESeq2 corrected for an FDR of 0.05) were considered differentially expressed. GO (biological process) pathways enriched or depleted in the different FUS mutant compared to the control motor neurons (using GSEA). For **d, e** in each Venn compartment, the top number represents upregulated genes/processes, while the bottom number represents downregulated genes/processes. Pathways with an adjusted p-value of < 0.05 were considered significantly up- or downregulated. **f–h** Heatmaps display the normalized enrichment scores (from GSEA) of selected groups of pathways from the Venn diagram in (**c**). Motor neurons are compared to V2a or other interneurons to show specific or stronger regulation of these pathways in motor neurons. **i** Top dysregulated pathways across FUS mutant motor neurons using the Ingenuity Pathways Analysis.

using CRISPR/Cas9-mediated genome editing (Fig. 3a). After verification of successful genome editing and that the mutation did not result in altered cellular TDP-43 protein levels in the iPSCs (Supplementary Fig. 6a–c), we conducted Smart-seq2 single cell RNA sequencing on motor neurons and interneurons (Fig. 3a) and combined it with our FUS data set. Following the same cell type classification (Supplementary Fig. 3e), we found that TARDBP M337V cells blended into the FUS dataset in UMAP projections (Fig. 3b, c). The localization of TDP-43 did not vary between control and TARDBP M337V motor neurons, similar to mouse models of the mutation, and neither was there a significant difference in phosphorylated TDP-43 levels[50,51] (Fig. 3d, e). The level of TARDBP mRNA remained stable across cell types and lines (Fig. 3f). DEA with DESeq2 demonstrated that the TARDBP M337V mutation induced a massive transcriptional response in motor neurons with 2284 DEGs, of which 2143 DEGs were unique to motor neurons and the majority upregulated. Interneurons showed only moderate responses with 145 DEGs for V2a and 196 DEGs for other interneurons, of which a minority (17 DEGs) were shared between the two interneuron groups (Fig. 3h, and Supplementary Data 3). We then identified potential upstream regulators which could explain the dysregulation elicited by mutant TARDBP, using the Ingenuity pathway analysis tools. Among the significant regulators that match the observed dysregulation (at a cutoff for the network bias corrected p-value < 0.001), we found TARDBP itself. We interrogated its functional status further by calculating its activation z-score. At −3.13, it indicates that a significant inhibition of TARDBP function occurs in TARDBP M337V motor neurons (Fig. 3i) despite its stable expression level. Collectively, we find that the TARDBP M337V mutation triggered a massive response in motor neurons, which was independent of protein mislocalization. Interneurons remained rather unaffected and thus, within this unique transcriptional dysregulation in motor neurons lies the key to their demise in ALS.

**Transcriptional changes shared across ALS-causative mutations in FUS, TARDBP and C9orf72 implicate mitochondrial dysfunction within motor neurons**

Disease histopathology appears unique with FUS[5,6] and TARDBP[18,52] mutations, but the selective motor neuron degeneration, glial activation and clinical presentation is similar, suggesting a convergence of pathomechanisms[53]. We reasoned that the comparison across FUS and TARDBP gene mutations could reveal if there are intrinsic pathways and DEGs central to motor neurons irrespective of the composition of aggregated proteins and disease etiology (Fig. 4a). We identified 1088 DEGs that were uniquely regulated in TARDBP M337V, but not in FUS motor neurons (Fig. 4b, Supplementary Data 4). There was considerable overlap between DEGs in TARDBP M337V and FUS P525L homozygous motor neurons, with 578 DEGs shared uniquely between the two lines (Fig. 4b, blue area). Furthermore, TARDBP M337V motor neurons shared 160 DEGs with FUS-ALS GOF (Fig. 4b, yellow area) and 64 DEGs with FUS-ALS LOF (Fig. 4b, red area). The GSEA between the TARDBP M337V and control motor neurons revealed 423 biological processes that were altered in the mutant motor neurons (Fig. 4c, Supplementary Data 5). Comparing mutant TARDBP M337V dysregulated processes with those found in mutant FUS motor neurons (Fig. 2d–h) showed that mutant TARDBP induced dysregulation in many more biological processes than mutant FUS, and the majority (370 biological processes) of these were thus unique to mutant TARDBP and showed a pattern of downregulation (217) (Fig. 4c). e.g., regulation of cholesterol biosynthetic process, ribonucleotide complex biogenesis, negative regulation of cellular amide metabolic process and ncRNA export from nucleus (Supplementary Fig. 6d). The majority of FUS dysregulated processes were also affected in TARDBP M337V motor neurons, with eight out of 11 FUS LOF (73%) processes and 45 out of 69 FUS GOF (65%) overlapping with GO terms in the mutant TARDBP motor neurons (Fig. 4c). This indicates that there is shared transcriptional dysregulation across the isogenic ALS lines that relate to early and common dysfunctions in motor neurons. Upon closer inspection, we find that the biological processes that were shared across mutant TARDBP and FUS LOF mostly relate to RNA metabolism (RNA splicing, mRNA processing, establishment of RNA localization), nuclear export and neuronal specific functions (dendritic spine development and neuron projection organization). The biological processes that were shared between mutant TARDBP and FUS GOF were upregulation of a number of RNA metabolic processes, e.g., RNA 3′ end processing, regulation of splicing, RNA localization, RNA export from nucleus, mRNA transport as well as downregulation of energy metabolic processes, e.g., oxidative phosphorylation and aerobic respiration (Fig. 4d). We also conducted an upstream regulator analysis on our full dataset to identify master regulators that agree with the observed differential expression in each motor neuron line (at a cutoff for the network bias corrected p-value of < 0.001, Supplementary Fig. 7a). We identified three significant protein-coding regulators shared across ALS-lines, which are all mitochondria-associated factors. The activation z-score for two of these factors CAB39L and DAP3 indicate their significant inhibition across the ALS lines (Supplementary Fig. 7b).

By comparing the motor neuron response across FUS and TARDBP mutations, we identified a shared and intrinsic signature of potentially early cellular dysfunction across ALS-causative mutations, which is characterized by changes in RNA metabolism, neuronal organization and mitochondrial mechanisms and may explain their particular susceptibility in ALS. To investigate if our findings were relevant more broadly across ALS, we compared our dataset with published bulk RNA sequencing from iPSC-derived cultures containing motor neurons that

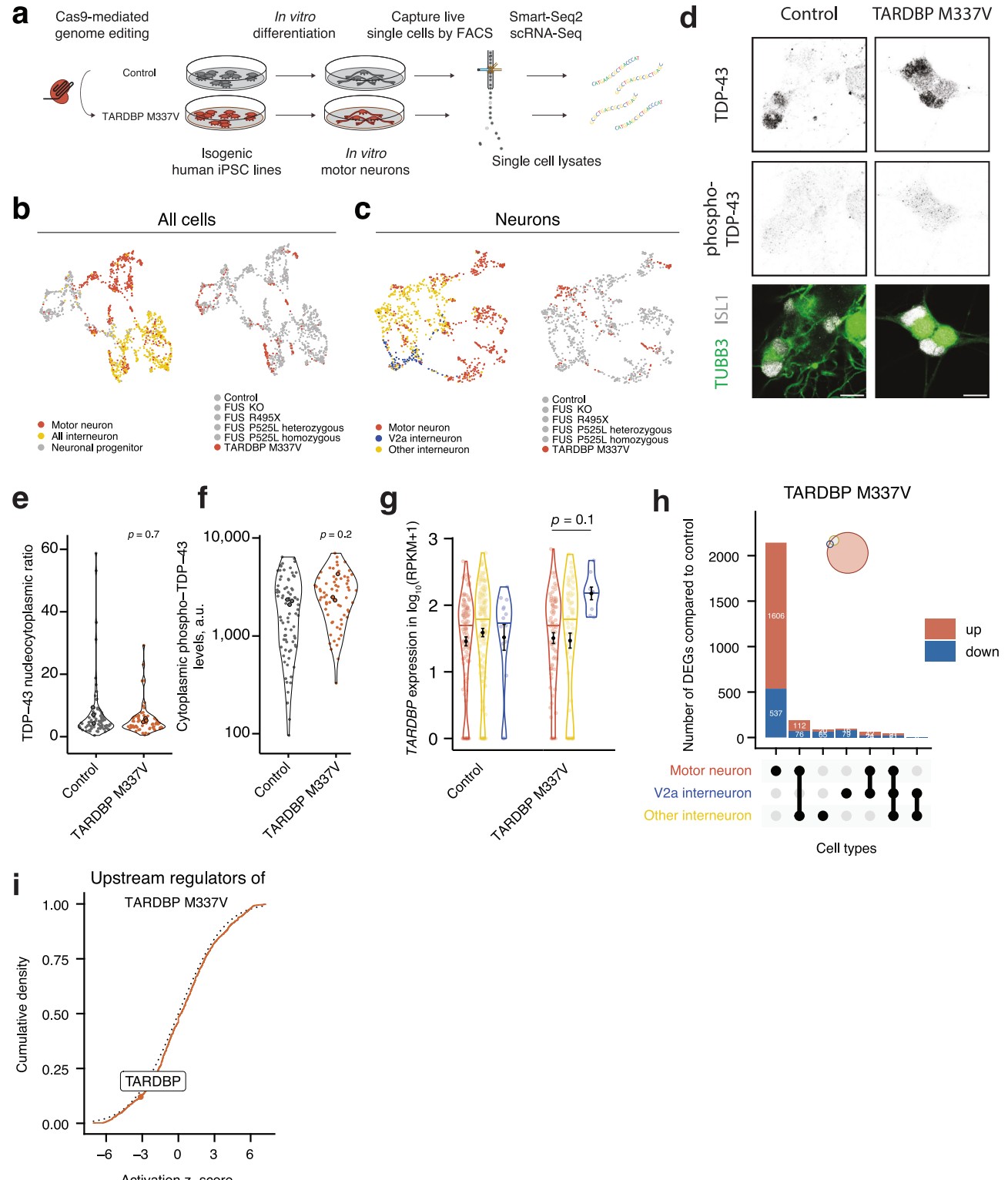

harbour ALS-causative *C9orf72* hexanucleotide repeat expansions (HRE). This dataset contained two *C9orf72*-ALS patient lines and corresponding corrected control lines, in which the HRE were excised[33]. Following DEA and GSEA, we identified 609 biological processes that were dysregulated with the *C9orf72* HRE. Comparison with our FUS-ALS sets revealed that 601 of these biological processes were uniquely regulated in the C9orf72 dataset. No FUS-ALS LOF processes were identified in C9orf72-ALS (Fig. 4e). However, the FUS-ALS GOF set shared eight biological processes with the C9orf72 mutant motor neurons, including downregulation of oxidative phosphorylation, upregulation of nuclear transport, establishment of organelle localization and negative regulation of the execution phase of apoptosis (Fig. 4e, f). The TARDBP M337V line shared an additional 41 dysregulated GO terms with the C9orf72 data set, demonstrating their higher similarity, compared to FUS-ALS, fitting with the TDP-43 neuropathology seen in *C9orf72*-ALS (Fig. 4e). Interestingly, as no TDP-43 protein pathology is visible in either *TARDBP* or *C9orf72* mutant motor neurons at this time point, these joint early disease mechanisms

**Fig. 3 | The TARDBP M337V mutation results in transcriptomic dysregulation unique to motor neurons. a** Schematic of the generation of the TARDBP M337V mutant iPSC line, differentiation into motor neurons and subsequent single-cell sorting and RNA-sequencing. **b** UMAP projections of all single cells in the study that passed quality control. The two UMAPs are colored by assigned cell type (left) or cell line (right). **c** UMAP projections of all neuronal cells in the study that passed quality control, again colored by assigned cell type (left) or cell line (right). **d** TDP-43 is mainly nuclear in the TARDBP M337V line. Confocal fluorescence microscopy with immunostaining of antibodies against TDP-43 and phospho-TDP-43 (S409/S410) in motor neurons (TUBB3 + , ISL1/2 + ) at day 28. Scale bars are 10 μm. **e**, **f** Violin plots for the quantitative immunofluorescence measurements of intracellular TDP-43 levels and phospho-TDP-43 levels in ISL1⁺ TUBB3+ motor neurons. The nucleocytoplasmic ratio for TDP-43 was calculated and shown in (**e**) and the cytoplasmic levels of phospho-TDP-43 (S409/S410) is shown in (**f**). The dots with black represent the average per independent differentiation ($n = 3$), while other dots represent individual well averages. The p-values were obtained from two-sided Wilcoxon rank sum test. **g** Violin plots of the *TARDBP* mRNA expression in the control and TARDBP M337V lines across cell types. The normalized expression is given as RPKM. Differential expression was performed within cell type, across lines. Error bars show mean ± SEM, * $p < 0.05$, ** $p < 0.01$, *** $p < 0.001$, derived from two-sided DEA testing using DESeq2 corrected for an FDR of 0.05. The p-value for the V2a to motor neuron comparison in the TARDBP M337V line is 0.11. **h** Upset plots showing the overlap between DEGs in motor neurons, V2a interneurons and other interneurons between the TARDBP M337V mutant lines versus the control cell line. Each column represents a section of a Venn diagram of the cell types as indicated in the table below. The corresponding Venn diagram is shown in an insert. The red top and blue bottom columns represent upregulated and downregulated genes, respectively. Genes with an adjusted p-value of <0.05 (two-sided DEA with DESeq2) were considered differentially expressed. **i** Analysis of upstream regulators in the DEGs between TARDBP M337V and control motor neurons using the Ingenuity Pathway Analysis suite. The regulator TARDBP is highlighted with an activation z-score of -3.13, indicative of its inhibition in the TARDBP M337V cell line.

---

appear independent of protein aggregation. Lastly, we also assessed the pairwise overlap of up- and downregulated pathways between each mutant ALS line (Fig. 4g). Here we used the $\log_2$ odds ratio of the intersection against all dysregulated pathways across lines, which is positive for an overlap larger than expected if pathways were selected randomly from all dysregulated pathways, or negative for an overlap smaller than expected. Overall, we find significant similarities in up- or downregulated pathways across ALS-mutant motor neurons with odds ratios significantly exceeding the expected value by chance selection. The upregulated pathways between the *FUS* and *TARDBP* ALS-mutant motor neurons are highly similar to each other as are the down regulated pathways. The upregulated pathways in FUS KO motor neurons are also highly similar to the upregulated pathways in the FUS R495X and either heterozygous or homozygous FUS P525L motor neurons, suggesting that some FUS LOF is indeed manifested in those lines. In contrast, we find significant dissimilarities in the upregulated pathways between our ALS-FUS or ALS-*TARDBP* motor neurons and the ALS-*C9orf72* motor neurons with negative $\log_2$ odds ratios for each pairwise comparison, while there is a higher degree of similarity in the down regulated pathways between these sets. Thus, the overall shared dysregulation across these ALS datasets lies mainly in the downregulated pathways in face of considerable overall dissimilarity between our ALS-*FUS* or ALS-*TARDBP* mutant lines and the ALS-*C9orf72* dataset in the pathways that are upregulated.

In conclusion, we find that *FUS* and *TARDBP* mutations trigger the dysregulation of many shared processes in motor neurons, several of which implicate dysfunctional mitochondrial energy metabolism. Comparison of our findings to motor neurons derived from *C9orf72*-ALS patient iPSCs on non-isogenic backgrounds, which can limit detectability, revealed early mitochondrial metabolic dysfunction as a broadly shared pathway across these three distinct genetic causations, strengthening the importance of our finding for ALS in general.

### Downregulation of key mitochondrial pathways across isogenic ALS lines points to metabolic impairment

Our transcriptomic analysis consistently identified biological processes associated with mitochondrial energy metabolism across in vitro-derived ALS mutant motor neurons. Notably, 14 downregulated biological processes of the 45 shared among the *FUS*-ALS GOF and *TARDBP*-ALS motor neurons were linked to mitochondria. Mitochondrial dysfunctions have also been recognized as hallmarks of ALS pathology in patients as well as animal and cellular models[33,54–58]. We therefore decided to assess mitochondrial processes and pathways in more detail. At closer inspection, 54 of 123 mitochondrial respiratory genes were dysregulated in motor neurons in at least one ALS line (FUS or TARDBP) compared to the control at a cut-off of $p_{adj} < 0.001$. Most of these genes were downregulated in ALS motor neurons and spanned respiratory complexes I–V (Fig. 5a). In GSEA, we looked into three

mitochondrial pathways that were specifically downregulated in ALS motor neurons (Fig. 5b, d, f) and the expression of selected genes within these (Fig. 5c, e, g). For the GO term "ATP synthesis-coupled electron transport" the associated genes were mostly downregulated across ALS mutations (Fig. 5b), which is illustrated by its members *COX6A1*, *NDUFS6*, and *NDUFA12* (Fig. 5c). For "Oxidative phosphorylation" (Fig. 5d), *ATP5MF*, *COX7C*, and *NDUFB9* were clearly downregulated (Fig. 5e). Among the 37 mitochondrially encoded genes, the majority were downregulated across ALS mutations (Fig. 5f), exemplified by *MT-CO3*, *MT-ND3*, and *MT-ATP6* (Fig. 5g). To confirm these differences in RNA expression (Supplementary Fig. 8) are also reflected on the protein level, we performed immunofluorescent staining against MT-CO1, MT-CO2, MT-CO3 and NDUFA12 in combination with staining against TOM22, as a marker of mitochondria, and neurofilament intermediate chain (NEFM) (Fig. 5h). Systemic quantification of these markers within mitochondria in motor axons demonstrated a reduction of mitochondrial MT-CO1 (Fig. 5i), and MT-CO2 (Fig. 5j) in the FUS R495X line. MT-CO3 was downregulated in mitochondria in TARDBP M337V motor axons (Fig. 5k), where it showed the largest decrease in mRNA level (Fig. 5g). Mitochondrial levels of the NDUFA12 protein were also reduced in the FUS R495X line (Fig. 5l), consistent with this line showing the largest downregulation at the mRNA level (Fig. 5c). This analysis highlights the various reduction of mitochondrial expression, that occur in ALS mutant motor neurons, across mutations and confirms several key findings also on the protein level in mitochondria in motor axons. Altogether the downregulation of these mitochondrial genes suggests that oxidative respiration is commonly compromised early on in in vitro-derived ALS motor neurons.

We then aimed to functionally assess cellular energy metabolism and mitochondrial activity in intact ALS motor neurons as oxygen consumption rate (OCR) and extracellular acidification rate (ECAR) using Seahorse XF e96 extracellular flux analyzer. On day 21 after seeding, a time-point at which motor neurons have become metabolically active, the neurons were measured while injecting inhibitors of mitochondrial function and glycolysis to dissect underlying molecular mechanisms (Supplementary Fig. 9a, b,c). These commonly used assays to address overall changes in cellular bioenergetics did not reveal significant differences in ATP-linked respiration, maximal substrate oxidation or coupling efficiency in our ALS lines (Supplementary Fig. 9d). Permeabilizing the cells to remove differences in substrate supply and to directly feed substrates into the mitochondrial respiratory chain showed no differences in Complex I, II or IV activity (Supplementary Fig. 9e). To exclude that these results were specific to the DF6-9-9T.B background, we confirmed the measurements with FUS R495X and TARDBP M337V lines on the KOLF2.1J background. Only the *TARDBP M337V* line showed a trend towards lower substrate oxidation (Supplementary Fig. 9f). Despite technical optimization efforts, including high density culturing to reach sufficient respiration rates,

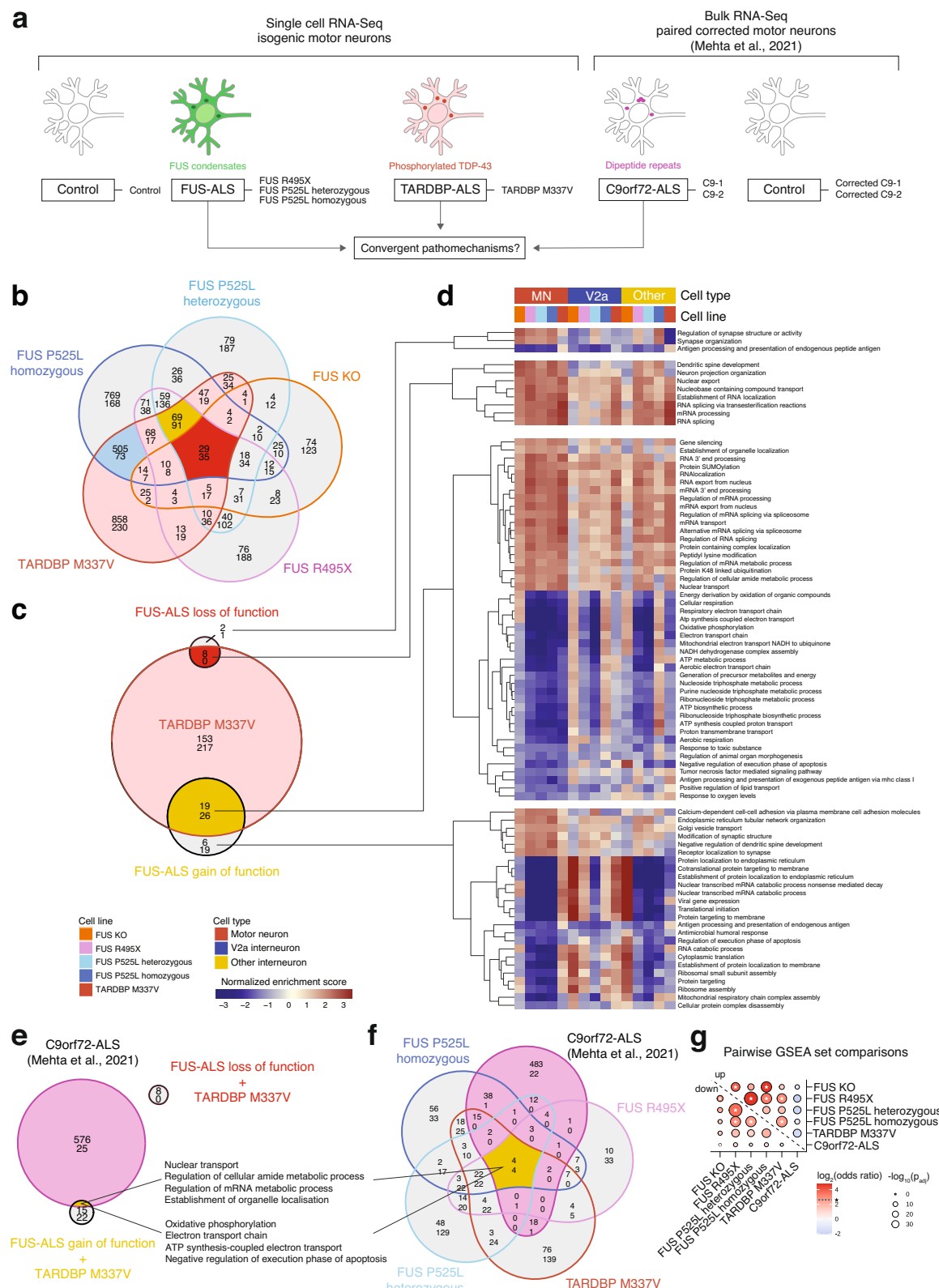

we noted high variability in the independent measurements. This is likely caused by the uneven distribution of cells (Supplementary Fig. 9h), which inevitably form clusters at the required cell densities. The incomplete coverage by the measurement sensor was previously reported to affect OCR measurements[59]. At this stage, neurons at later time-points than day 28 could not be measured as these detached during preparation (Supplementary Fig. 9h). We conclude that the

dynamic nature of the high-density cultures, which results in uneven distribution and detachment of cell clusters, is a limiting factor of this experimental setup, requiring further technological improvements.

We also quantified the copy numbers of mitochondrial genomes by qPCR in our ALS-motor neuron over time (Fig. 5m,n) and applied a multiple linear regression model accounting for cell numbers and differentiation round (the main sources of experimental of variability).

**Fig. 4 | Cross-comparisons among FUS-, TARDBP-, and *C9orf72*-ALS reveal shared dysfunctions that converge on metabolic pathways. a** Strategy to uncover shared dysfunction across familial ALS forms that could uncover convergent pathomechanisms that are intrinsic to motor neurons. Cross-comparison of *FUS*-ALS and *TARDBP*-ALS motor neurons across our single-cell RNA-Seq data set (**b**–**d**) followed by cross-comparison with external bulk RNA-Seq dataset from *C9orf72*-ALS[33] was thus conducted (**e**, **f**). **b** Venn diagram showing overlap of DEGs between all FUS mutant motor neurons, as well as the TARDBP mutant motor neurons, as compared to control. Genes with an adjusted *p*-value of <0.05 were considered differentially expressed. **c**, Venn diagram showing overlap in dysregulated biological processes between FUS LOF lines (all FUS lines), FUS GOF lines (excluding the FUS KO), and the TDP-43 M337V line following GSEA. For **b**, **c** in each compartment, the top number represents upregulated genes/pathways, while the bottom number represents downregulated genes/pathways. Pathways with an adjusted *p*-value of <0.05 (two-sided DEA with DESeq2 corrected for an FDR of

0.05) were considered significantly up- or downregulated. **d** Heatmaps displaying normalized enrichment scores from GSEA of selected groups of pathways from the Venn diagram in (**c**). Motor neurons are compared to V2a interneurons to show specific or stronger regulation of these pathways in motor neurons. **e** Venn diagram depicting the overlap between dysregulated biological processes in TARDBP M337V and the ALS-FUS GOF and LOF groups defined in Fig. 3. **f** Venn diagram depicting the overlap between dysregulated pathways from the single cells RNA-Seq of ALS mutant motor neurons (FUS R495X, FUS P525L lines and TARDBP M337V) and the bulk RNA-Seq of *C9orf72*-ALS motor neurons. **g** Pairwise set comparisons of the dysregulated pathways in the Venn diagram from (**f**). The odds ratio of the overlap against all dysregulated pathways in the Venn diagram is shown as the bubble colour. The bubble size is the adjusted *p*-value of the overlap calculated by two-sided Fisher's exact test; the asterisk indicates comparisons for which the odds ratio is larger than five-fold.

In this approach we observe a small significant effect on mitochondria in the FUS P525L homozygous motor neurons ($p = 0.035$), which also significantly interacts with time-points.

## Impaired mitochondrial motility is seen across ALS mutant motor axons

Next, we decided to investigate mitochondrial distribution dynamics in axons as a possible readout of mitochondrial function, which may not yet impact metabolic function. Frequently spaced mitochondria are required for normal axonal function, especially electrical signal transduction. The axonal transcriptome of motor neurons is consequently highly enriched in genes involved in the mitochondrial respiratory chain compared to the soma[37]. By seeding embryoid bodies with motor neuron progenitors at day 14, followed by terminal differentiation and maturation during which motor axons grow radially and continuously outwards. In these cultures, axons can grow to more than 2 mm in length by day 28 (Fig. 6a). To assess mitochondrial motility in these axons, we imaged live neurons following a staining with a mitochondrial tracking dye (Fig. 6b–d). We assessed the movement of mitochondria along the anterograde-retrograde axonal axis in live cell time lapse microscopy (Fig. 6e), by recording axons during six min time lapses at 0.5 Hz and tracking mitochondria using TrackMate 7[60]. We tracked 14,855,019 spots, total number of mitochondria in all frames, in this manner and assembled 459,479 individual track branches (total mitochondria over the time-lapses). We classified mitochondria into stationary/oscillating mitochondria, mitochondria that move in anterograde or retrograde direction, while we discarded stray movements orthogonal to the axons. To be able to discern if mitochondrial movement was linked to protein mislocalization or not, we included the FUS R244C line, which does not show protein mislocalization (Fig. 2a–c). We found that all ALS motor neurons displayed an increased number of stationary mitochondria compared to control motor neurons, which was also reflected in a corresponding decrease of motile mitochondria in the anterograde and retrograde direction (Fig. 6f). The change in mitochondrial motility was significantly associated with the mutations (Pearson's $\chi^2$ test, $\chi^2 = 88192$, df = 12, $p < 2.2e\text{-}16$). We found that stationary mitochondria were strongly overrepresented in all ALS motor neurons, while they were underrepresented among control and *FUS* KO motor neurons (Fig. 6g). This analysis of mitochondrial motility indicates further disturbance of mitochondria across ALS lines, which are linked to GOF toxicity. As we see the dysfunction in mitochondrial motility also in FUS R244C motor neurons, this early defect appears independent of protein mislocalization. Dysfunctional mitochondrial motility can impair the neuron's ability to properly spread mitochondria across the motor axon as well as to repair and turn over mitochondria normally. We thus measured 1) the distance between individual mitochondria and 2) the distance between the adjacent anterograde and retrograde neighbours for each mitochondrion to further assess their

axonal distribution. Indeed, we observed a significantly increased spreading of mitochondria in all ALS motor axons compared to the control line using either measure of mitochondrial spacing (Fig.6h). The largest differences were observed in the FUS R495X and TARDBP M337V motor axons, in which mitochondria were spaced out approximately twice as far as the control motor neurons with an average distance of 14.8 μm and 11.4 μm from the retrograde to anterograde adjacent neighbour in contrast to 6.0 μm control motor axons (Fig. 6h). We also interrogated how regular mitochondria are spaced out by calculating the difference of the distance from each mitochondrion to its anterograde and retrograde neighbor and found significant deviations in the distribution of mitochondria in the ALS motor axons, indicative of an overall more irregular spacing of mitochondria (Fig. 6h). Collectively, in line with our transcriptomic data, we identify reduced mitochondrial motility and increased spacing of mitochondria in ALS motor neurons across *FUS* and *TARDBP* mutants, independent of protein mislocalization.

## Discussion
Motor neurons are highly vulnerable to degeneration in ALS and show early loss of synapses with skeletal muscle[61]. In order to understand the initial intrinsic mechanisms that could lead to their early demise across disease etiologies, we compared the impact of ALS mutations in *FUS* and *TARDBP* on the transcriptomes of in vitro-derived spinal motor neurons and interneurons at single cell resolution. Isogenic lines allowed us to detect subtle transcriptomic changes that could easily have been masked in lines with diverse genetic backgrounds, even if compared with corrected controls for each non-isogenic line. Furthermore, the generation of lines harbouring either FUS NLS (R945X and P525L) or FUS non-NLS (R244C) abrogating mutants allowed us to discern pathomechanisms entirely uncoupled from innate FUS mislocalization. Motor neurons responded to *FUS* and *TARDBP* mutations with a much greater transcriptional dysregulation than interneurons, which mirrors their higher susceptibility to intrinsic pathological mechanisms in ALS. Our findings in *FUS* and *TARDBP* lines are complemented and corroborated by similar results in neurons specified from familial *C9orf72*- and sporadic ALS patient iPSCs that demonstrated a lesser transcriptional response in V2a and V1 Renshaw interneurons compared to susceptible motor neurons[62]. While interneurons are more resilient to ALS than most motor neurons in vivo, glutamatergic V2a interneurons in the brainstem and spinal cord also degenerate, but only at late stages of the disease, which aggravates respiratory failure[41]. It is therefore likely that our analysis detects an early dysfunction in motor neurons specifically that could be at the heart of why they display higher intrinsic vulnerability in response to ALS mutations.

In motor neurons, we observed that individual *FUS* mutations elicit distinct, but also shared transcriptomic signatures. Cross comparison of FUS R495X and P525L with FUS KO motor neurons allowed

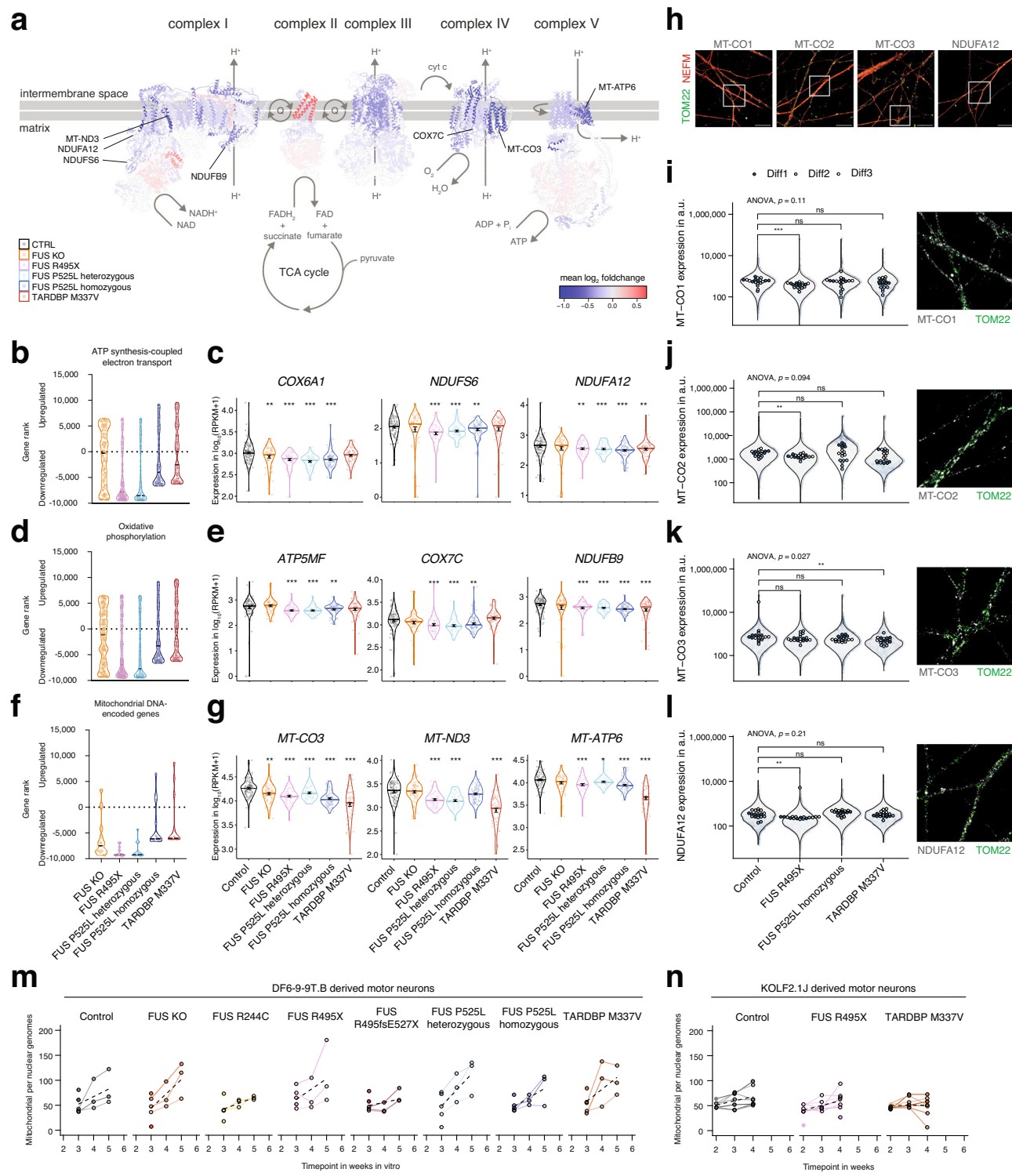

us to dissect the contribution of LOF and GOF in FUS to this shared transcriptional signature, the majority of which is associated with a potentially toxic GOF. The shared GOF might arise due to the mis-localization of FUS, given that the R495X and P525L mutations ablate FUS' NLS, resulting in its accumulation in the cytoplasm, as we also observe in our motor neurons. Cytoplasmic mutant FUS displays toxic GOF properties across cellular models, including the sequestration of various RNA species and proteins[5,6,12]. Although we only detect few ALS-FUS linked LOF pathways, they mirror defects in dendrites in rodents with FUS depletion or overexpression of ALS mutations[63–65] and potentially compensatory changes in splicing regulation because

FUS is a non-essential factor in neurogenesis after all[66]. While the loss of FUS does not elicit motor neuron disease on its own[66] the interplay of GOF and LOF associated changes in *FUS*-ALS might exacerbate the course of the disease as in late stages the abundant cytoplasmic FUS aggregates also sequester nuclear FUS (two-hit hypothesis)[5,67,68] So far little is known about the contribution of these processes across different cell types, let alone spinal motor neurons. Our analysis of mutant FUS motor neurons affirms that LOF changes are indeed less pronounced than GOF changes. This is potentially due to the fact that the FUS protein belongs to the FET family of proteins that share similar functions and hence EWS and TAF15 might partially compensate for

**Fig. 5 | Dysregulation in key mitochondrial genes and pathways points to mitochondrial impairment in motor neurons across isogenic ALS lines. a** The mean expression changes of respiratory genes in ALS motor neurons (as mean log$_2$ foldchange of FUS R495X, FUS P525L heterozygous, FUS P525L homozygous, and TARDBP M337V compared to control) were mapped to the protein structures of the mammalian mitochondrial respiratory complexes I–V (PDB entries 5lc5, 1zoy, 1bgy, 1occ, and 5ara) and rendered in UCSF Chimera. The composition was inspired by Sousa et al. The location of genes from expression plots in (**c, e, g**) are indicated with pointers. **b, d, f** Modified GSEA plots showing the rank of all genes within this specific GO-term for all mutant motor neurons generated using the homogeneous protocol, according to the GSEA test statistic. The most significantly upregulated gene is on the top and the most significantly downregulated gene on the bottom. **c, e, g** Violin plots showing normalized expression values (RPKM) of three example genes of the pathway from (**b, d, f**) in motor neurons generated with the homogeneous protocol. Error bars show mean ± SEM. * $p < 0.05$, ** $p < 0.01$, *** $p < 0.001$ vs. control, derived from two-sided DEA testing in DESeq2 corrected for an FDR of 0.05. The p-values for COX6A1 are 9.5E-3 (FUS KO), 4.3E-7 (FUS R495X), 6.4E-10 (FUS P525L heterozygous), 2.7E-7 (FUS P525L homozygous), and 6.8E-2 (n.s., TARDBP M337V). The p-values for NDUFS6 are 9.1E-1 (n.s., FUS KO), 5.3E-5 (FUS R495X), 1.7E-4 (FUS P525L heterozygous), 5.5E-3 (FUS P525L homozygous), and 3.4E-1 (n.s., TARDBP M337V). The p-values for NDUFA12 are 6.6E-1 (n.s., FUS KO), 4.9E-3 (FUS R495X), 4.4E-4 (FUS P525L heterozygous), 4.5E-6 (FUS P525L homozygous), and 2.6E-3 (TARDBP M337V). The p-values for ATP5MF are 7.8E-1 (n.s., FUS KO), 2.7E-6 (FUS R495X), 4.6E-5 (FUS P525L heterozygous), 1.2E-3 (FUS P525L homozygous), and 2.1E-1 (n.s., TARDBP M337V). The p-values for COX7C are 9.6E-1 (n.s., FUS KO), 1.5E-5 (FUS R495X), 7.6E-6 (FUS P525L heterozygous), 2.0E-3 (FUS P525L homozygous), and 6.1E-1 (n.s., TARDBP M337V). The p-values for NDUFB9 are 3.5E-1 (n.s., FUS KO), 1.3E-4 (FUS R495X), 3.8E-6 (FUS P525L heterozygous), 1.2E-8 (FUS P525L homozygous), and 1.2E-4 (TARDBP M337V).The p-values

for MT-CO3 are 2.5E-3 (FUS KO), 2.4E-7 (FUS R495X), 8.5E-4 (FUS P525L heterozygous), 5.1E-9 (FUS P525L homozygous), and 6.0E-7 (TARDBP M337V). The p-values for MT-ND3 are 7.5E-1 (n.s., FUS KO), 5.3E-9 (FUS R495X), 6.2E-10 (FUS P525L heterozygous), 1.8E-1 (n.s., FUS P525L homozygous), and 1.2E-11 (TARDBP M337V). The p-values for MT-ATP6 are 5.3E-1 (n.s., FUS KO), 3.2E-4 (FUS R495X), 1.9E-2 (FUS P525L heterozygous), 3.3E-5 (FUS P525L homozygous), and 1.4E-13 (TARDBP M337V). **h** Confocal fluorescence microscopy with immunostaining of mitochondrial complex I (NDUFA12) or complex IV (MT-CO1, MT-CO2, MT-CO3) markers was performed in several wells per motor neuron line and in three independent differentiations each. The axons and mitochondria were stained using NEFM and TOM22 markers respectively. Scale bars are 30 μm. The location of example images for panels i-l are indicated by boxes. **i–l** Quantitative immunofluorescence of the expression levels of MT-CO1, MT-CO2, MT-CO3, and NDUFA12 in TOM22⁺ mitochondria in NEFM⁺ neurites. The violin plots represent the expression levels in individual mitochondria, the dots are the mean in replicate regions, and the colors are three independent motor neuron differentiation with the homogenous protocol. The p-values were calculated with two-sided Wilcoxon rank sum test between mutant and control motor neurons and significance levels are ns $p >= 0.05$, * $p < 0.05$, ** $p < 0.01$, *** $p < 0.001$. The p-values for *MT-CO1* expressions are 0.0023 (FUS R495X), 0.49 (FUS P525L homozygous), 0.19 (TARDBP M337V). The p-values for *MT-CO2* expressions are 0.0014 (FUS R495X), 0.94 (FUS P525L homozygous), 0.085 (TARDBP M337V). The p-values for *MT-CO3* expressions are 0.12 (FUS R495X), 0.064 (FUS P525L homozygous), 0.0014 (TARDBP M337V). The p-values for *NDUFA12* expressions are 0.0013 (FUS R495X), 0.15 (FUS P525L homozygous), 0.89 (TARDBP M337V). **m, n** Mitochondria in motor neurons were quantified by the copy number of mitochondrial genomes per nuclear genomes by qRT-PCR on total DNA extracted from cultures at the indicated time-points in motor neurons from the DF6-9-9T.B (**m**, our lines) and KOLF2.1 J background (**l**).

---

the loss of FUS in the context of nuclear depletion[69,70]. The few shared dysfunctions we identified across *TARDBP, FUS* and *C9orf72* datasets belonged only to GOF. In ALS patient tissue, *C9orf72*- and *TARDBP*-ALS converge on cytoplasmic TDP-43 aggregation and thus it may be anticipated there would be greater convergence across these lines than for *TARDBP*- and *FUS*-ALS. When aggregation starts is unknown, but it relies on other dysfunctions that develop over time. *C9orf72* and *TARDBP* mutant motor neurons derived from iPSCs do not show obvious signs of TDP-43 mislocalization as they are in a very early phase of pathology. Nevertheless, early dysfunction in multiple cellular processes is apparent in such neurons[27,33,48,71,72], which highlights that TDP-43 cytoplasmic aggregation is not an early event in ALS. Neither is it required for degeneration as shown in model systems[22,23]. Thus, convergence seems to occur early in disease, preceding obvious TDP-43 cytoplasmic mislocalization. Furthermore, as the *C9orf72*-ALS patient lines used have a different genetic background compared to our isogenic lines, is possible that subtle converging dysfunctions that could have been detected in isogenicity are lost, particularly as differences between individuals are known to drive variation in iPSCs, both in differentiation capacity and morphology[73].

Our cross-comparison with the bulk sequencing data from *C9orf72*-ALS patient derived motor neurons reveals unique responses, but also further narrowed the signature of dysregulated genes that we detected in mutant *FUS* and *TARDBP* motor neurons. This signature contains transcriptional changes that are broadly shared among familial ALS motor neurons derived in vitro and might contain early converging pathological pathways. In this group, we identified the ablation of mitochondrial respiration as an early and broad pathological stage across disease causations. Mitochondria are central organelles for oxidative phosphorylation, lipid biogenesis, calcium homeostasis, and apoptosis. Their proper function and spacing is crucial in motor neurons with their exceptionally high demands for local energy supply and calcium buffering during electric signal transduction. Dysfunctional mitochondria are a hallmark of ALS found in end stage patient tissues[74,75]. Perturbations found across ALS model systems encompass changes in energy metabolism, mitochondrial

motility and morphology[58]. In ALS motor neurons derived from familial ALS iPSC lines with *SOD1*, *TARDBP* and *C9orf72* mutations or sporadic ALS iPSCs mitochondrial respiration was impaired, but not in FUS patient lines investigated so far[31,33,76,77]. However, in a longitudinal in vivo study using the FUS Δ14 mouse model, mitochondrial dysregulation was detected at onset of symptoms at age of 12 months but not earlier in bulk spinal cord samples[78]. Given that we detect signatures of mitochondrial dysfunction in our transcriptomic analysis at single cell resolution one could infer that mitochondrial dysfunction begins much earlier, before onset of other symptoms. However, technical barriers prevented us to establish whether this early transcriptomic signature already translates into functional mitochondrial respiration deficits.

There is also some evidence that *FUS* and *TARDBP* can both directly modulate mitochondria in ALS. Cytoplasmic FUS condensates were highly enriched for mitochondrial proteins[79], but FUS mislocalization can also indirectly interfere with mitochondria by sequestering crucial mitochondrial components e.g., such as DHX30 required for translation of mitochondrial mRNAs, which hampered the renewal of the respiratory complexes[80]. A small fraction of cellular TDP-43 localizes to mitochondria and the prevention of the mitochondrial localization of TARDBP M337V in mice ablated mitochondrial dysfunction[81]. Due to its partial mitochondrial localization, TDP-43 protein can interact with multiple crucial mitochondrial proteins and mRNAs, including prohibitin-2 (PHB2), VDAC1[82], and *MT-ND3* and *MT-ND6* mRNAs[56]. PHB2 is important for mitochondria in neurons and its loss caused mitochondrial instability and neurodegeneration[83,84].

While mitochondrial respiration could not be robustly investigated, we found a significant and early reduction in motile mitochondria in distal axons among our *FUS* and *TARDBP* ALS lines, in agreement with previous findings[32,33,71,85]. Indeed, mislocalized proteins in ALS can have a direct impact on axonal transport by binding to motor proteins and the axonal cytoskeleton that could explain part of the motility phenotype. For example, cytoplasmic FUS associates directly with both myosin Va[86] and KIF5[87] but also regulates the mRNA of several motor proteins, including *KIF5C, KIF1B, KIF3A*[88]. Likewise,

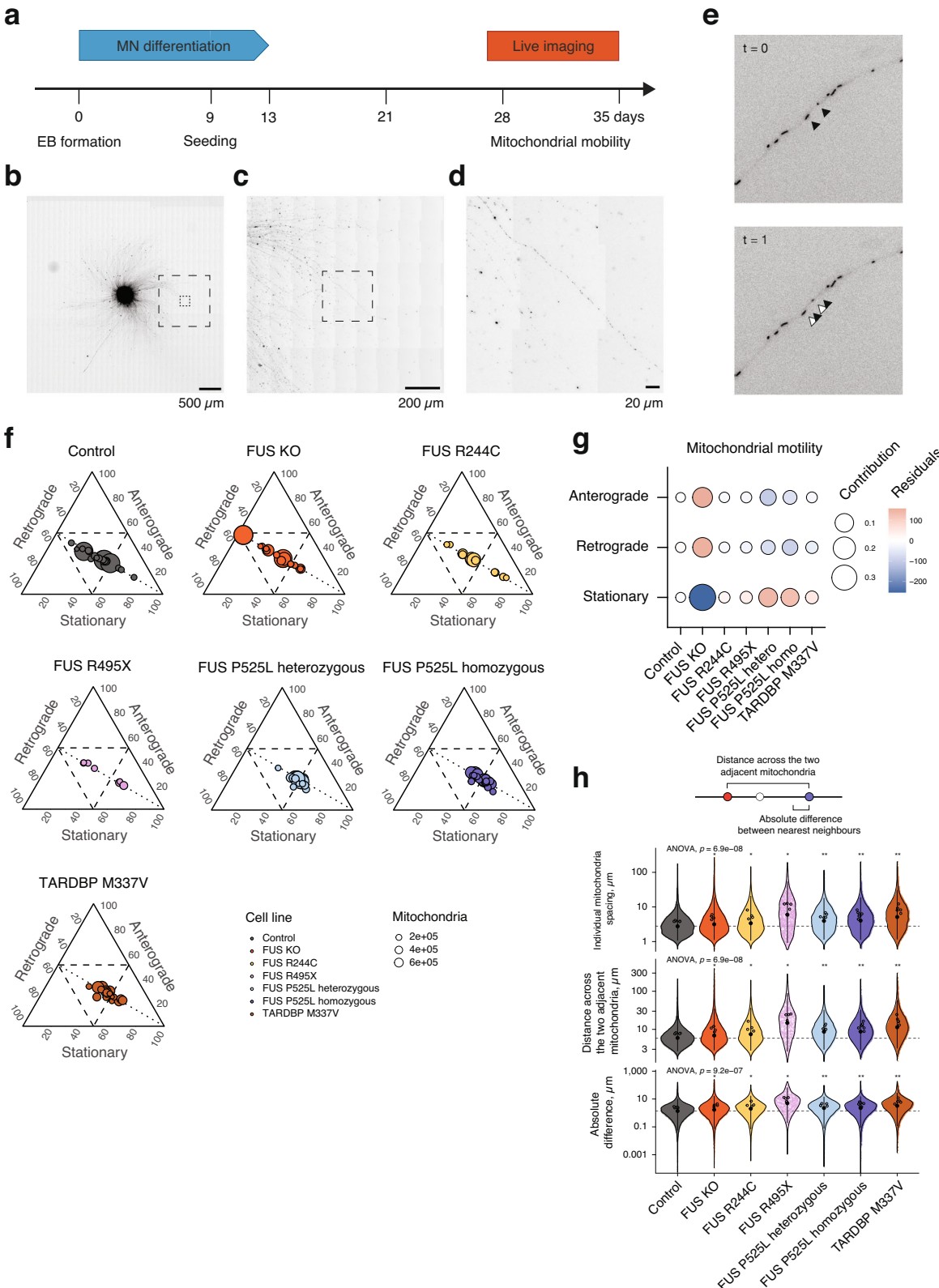

TDP-43 binds to various cytoskeletal proteins and regulates the *MAP1B* and *NEFL* mRNAs[89], important components of the neuronal cytoskeleton. Furthermore *C9orf72*-derived dipeptide repeat proteins can road-block transport on microtubules resulting in arrested mitochondria[90]. However, we also observed impaired trafficking of mitochondria in FUS R244C motor neurons, which indicates that mislocalization of FUS is not required for the motility defect. The

R244C mutation is defective in DNA repair function of FUS[91,92], and earlier work in iPSC-derived motor neurons suggests that nuclear genomic stress can act upstream of FUS aggregation and exacerbate axonal mitochondrial dysfunction[82,85] and represent mitochondrial replication defects found also in ALS patients[93]. The FUS mutations that we used also differ in their RNA binding ability due to their disposition of RGG domains which also modulate their ability for liquid-

**Fig. 6 | ALS motor axons show dysfunction in mitochondrial motility.**
**a** Schematic of the mitochondrial motility assays using live fluorescence micro-scopy. Motor neuron progenitors were attached as neurospheres, followed by terminal differentiation and axon outgrowth. Mitochondria were labeled using TMRM. **b**–**d** Individual mitochondria become discernable at increasing magnifica-tions (**b**, **c**), and were recorded in peripheral axons (**d**). Scale bars are as indicated, ranging from 500 μm down to 20 μm. **e** 6-min long timelapses were recorded at 0.5 Hz in 3,4 locations in at least 3 attached neurospheres from several independent differentiations per line (namely control $n = 4$, FUS KO $n = 5$, FUS R244C $n = 3$, FUS R495X $n = 2$, FUS P525L heterozygous $n = 4$, FUS P525L homozygous $n = 5$, TARDBP M337V $n = 5$). Traces were identified using TrackMate 7 with StarDist detector and simple LAP tracer and directional movement was analyzed in R using our package 'mitotrackR'. **f** The proportions of stationary and motile mitochondria were cal-culated. The three axes represent these proportions in the three motility groups in ternary plots. Each bubble represents the number of mitochondria in individual recordings. The dashed lines indicate 50% thresholds in each motility group and the dotted lines are the anterograde-retrograde isoproportional lines (at which the anterograde and retrograde movement are equal). **g** The influence of the mutation in isogenic motor neurons on the mitochondrial motility (as the proportions of stationary, anterograde and retrograde mobile mitochondria) was assessed by Pearson's $\chi^2$ test (one-sided, $\chi^2 = 88192$, df = 12, $p < 2.2e\text{-}16$). The association between isogenic mutants and mitochondrial motility is plotted. The sign of the

residuals indicates the direction of the association, with attraction and repulsion colored as red and blue, respectively. The bubble size represents the contribution to the total $\chi^2$ score (squared residuals over $\chi^2$ statistic). **h** Mitochondria align across motor axons, such that each mitochondrion has a pair of nearest neighbors in the anterograde and retrograde direction. In the first frame of each recording from (**f**, **g**) the Euclidean distances between individual mitochondria as well as the dis-tance from each mitochondrion to the pair of adjacent mitochondria was measured to then calculate the distance across the three mitochondria as well as the absolute difference between the pair of nearest neighbors in the opposing directions. The violin plots represent values for individual mitochondria, the colored dots the average value in each video, and the black dots and whiskers are median ± the confidence interval ($\alpha = 0.05$). ANOVA was followed by *post-hoc* two-sided Wil-coxon rank sum test between mutant and control lines and the significance levels for the $p$-values are ns $p \geq 0.05$, * $p < 0.05$, ** $p < 0.01$, *** $p < 0.001$. The $p$-values for mitochondrial spacing are 0.0159 (FUS KO), 0.0159 (FUS R244C), 0.0159 (FUS R495X), 0.0079 (FUS P525L heterozygous), 0.0025 (FUS P525L homozygous), and 0.0079 (TARDBP M337V). The $p$-values for the distances across adjacent mito-chondria are 0.0159 (FUS KO), 0.0159 (FUS R244C), 0.0159 (FUS R495X), 0.0079 (FUS P525L heterozygous), 0.0025 (FUS P525L homozygous), and 0.0079 (TARDBP M337V). The $p$-values for the absolute difference are 0.0317 (FUS KO), 0.0317 (FUS R244C), 0.0159 (FUS R495X), 0.0079 (FUS P525L heterozygous), 0.0051 (FUS P525L homozygous), and 0.0079 (TARDBP M337V).

liquid phase transition as well as their oligomerisation potential[44]. Given that we observe reduced mitochondrial motility both in the presence and absence of cytoplasmic FUS as well as in the absence of apparent cytoplasmic TDP-43 mislocalization, our data suggests that mislocalization is not a prerequisite but suggests that different mole-cular mechanisms can direct mitochondrial motility defects.

Altogether the combined effects we observe could become det-rimental to motor neurons over time. Motor neurons are exceptionally vulnerable to energy depletion. One of the main reasons for this is that motor neurons are highly dependent on transport along their long axons, which can span over a meter to reach muscle targets in arms, legs, feet and hands. Furthermore, synapses such as the neuromus-cular junction are major sites of neuronal energy consumption and the trafficking of mitochondria to the synapses is thus critical to meet energy requirements of signaling[90,94]. Indeed, it has been estimated that $400\text{-}800 \times 10^6$ ATP molecules are required just to restore the electrochemical gradient of a neuron after a single action potential via $Na^+/K^+$ pumping[95–98]. Mitochondria have been estimated to generate 93% of the ATP in presynaptic terminals[99] and synapse physiology is clearly affected by local mitochondrial ATP production as local inhi-bition of oxidative phosphorylation or depletion of mitochondria here results in defective mobilization of synaptic vesicles, and failed neurotransmission[100–102]. Thus, the defects we find in mitochondrial transcriptome and target proteins could quickly lead to severe dys-function and miscommunication with muscles and weakening of synapses. Concomitantly, it was recently shown that targeting mito-chondria with a small molecule could improve upper motor neuron health in a TDP-43 mouse model[103]. The inability of ALS motor neurons to respond to high energy drains and the lack of compensatory gly-colysis could render the motor neurons intrinsically more vulnerable as well as more dependent on glial support[104,105].

Our data implicate mitochondria as important early targets in lower motor neurons. It has been shown that mitochondrial transport along axons is defective in iPSCs models of familial ALS[32,33,71], but it has not been investigated what regulates this disturbance and how it affects the neuromuscular junction and communication between motor neurons and muscle. ATP is also thought to be used as a hydrotrope to help solubilize hydrophobic proteins. Thus, ATP con-centrations could influence processes such as protein aggregation or liquid-liquid phase separation[106], events that both occur in ALS and are thought to be detrimental to motor neurons at late stages of the disease.

In conclusion, we find an intrinsic and early transcriptomic sig-nature of dysfunction across ALS motor neurons derived from human iPSCs along with less motile mitochondria implicating exhaustion of axonal energy supply and decline of proper function over time. Intri-guingly, we observe these perturbations also in the absence of appar-ent FUS and TDP-43 mislocalization, pointing towards mitochondria as an early and pivotal target to modulate in ALS.

## Methods
### Antibodies
Custom mouse monoclonal antibodies against the FUS C-termini were generated at Boster Bio (Pleasanton, CA, USA) with antigens of wild-type P525 (N-DSRGEHRQDRRERPY-C; clone P3-A6C4) or mutant P525L (N-CGKMDSRGEHRQDRRERLY-C; clone P1-E7B6). The custom rabbit anti-FUS (N-terminus) antibody was described[107]. The custom rabbit anti-CPSF2 (CPSF-100) was described[34]. Other antibodies used in this study were: mouse monoclonal anti-FUS 4H11 [Santa Cruz Bio-technology, sc-47711], mouse anti-GAPDH 6C5 [Santa Cruz Bio-technology, sc-32233], rabbit anti-TARDBP [Proteintech Group, 12892-1-AP], mouse anti-TARDBP clone 41-7.1 [Santa-Cruz Biotechnologies, sc-100871], rat anti-TARDBP (phospho-S409/S410) clone 1D3 [Sigma-Aldrich, MABN14], mouse anti-OCT3/4 [Santa Cruz Biotechnology, sc-5279], rabbit anti-NANOG H155 [Santa Cruz Biotechnology, sc-33759], mouse anti-TUBB3 clone Tuj1 [Biolegend, 801202], chicken anti-NFH polyclonal IgY [Millipore, AB5539], mouse anti-ISL1/2 clone 39.4D5 [DSHB, 39.4D5-c], rabbit anti-ISL1 [Abcam, ab20670], and mouse anti-Hb9 (MNX1) clone 81.5C10 [DSHB, 81.5C10-c], mouse monoclonal anti-TOM22 clone 1C9-2 [Sigma-Aldrich T6319-.2 ML], rabbit monoclonal anti-MT-CO1 clone EPR19642 [Abcam, ab203917], rabbit polyclonal anti-MT-CO2 [Proteintech 55070-1-AP], rabbit polyclonal anti-MT-CO3 [Sino Biological 202683-T10-100], rabbit polyclonal anti-NDUFA12 [Thermo Scientific PA5-58973] and rabbit polyclonal anti-GAD67 [Sigma ZRB1090].

Secondary antibodies were donkey anti-rabbit IgG IRDye 800CW [Li-Cor, 926-32213], donkey anti-mouse IgG IRDye 800CW [Li-Cor, 926-32212], donkey anti-rabbit IgG IRDye 680LT [Li-Cor, 926-68023], donkey anti-mouse IgG IRDye 680LT [Li-Cor, 926-68022], donkey anti-mouse IgG AlexaFluor 488 [Invitrogen, A-21202], donkey anti-rat IgG AlexaFluor 488 [Invitrogen, A48269 and A21208], donkey anti-rabbit IgG AlexaFluor 546 [Invitrogen, A10040], donkey anti-rabbit IgG AlexaFluor 647 [Invitrogen, A31573], donkey anti-chicken IgY Alexa-Fluor 647 [Invitrogen, A78952], goat anti-rabbit IgG AlexaFluor 488

[Invitrogen, A21206], goat anti-mouse IgG2a AlexaFluor 488 [Invitrogen, A21131], goat anti-mouse IgG2b AlexaFluor 488 [Invitrogen, A21141], goat anti-mouse IgG1 AlexaFluor 568 [Invitrogen, A21124], goat-anti-mouse IgG2a AlexaFluor 568 [Invitrogen, A21134], goat anti-mouse IgG2a AlexaFluor 647 [Invitrogen, A21241], goat anti-mouse IgG2b AlexaFluor 647 [Invitrogen, A21242], and goat anti-rabbit IgG AlexaFluor 405 [Invitrogen, A31556].

## Plasmids

pCRISPR-EF1a-eSpCas9(1.1) has been previously described[34]. To generate the pCRISPR-EF1a-eSpCas9(1.1)-R495 plasmids two complementary oligonucleotides (sre326: 5'-CAC CGG GAC CGT GGA GGC TTC CGA-3', sre327: 5'-AAA CTC GGA GGC TCC AGG TCC C-3') or (sre324: 5'-CAC CGG ACC GTG GAG GCT TCC GAG-3', sre325: 5'-AAA CCT CGG AAG CCT CCA CGG TCC-3') and for M337 (JBA1: 5'-CAC CGC AGC ACT ACA GAG CAG TTG-3', and JBA2: 5'-AAA CCA ACT GCT CTG TAG TGC TGC-3') were subsequently phosphorylated using T4 polynucleotide kinase (PNK), annealed by heating to 95 °C and cooling to room temperature creating 5'-overhangs on both ends that were used to ligate into the BbsI sites of pCRISPR-EF1a-eSpCas9(1.1). The plasmid to express gRNA and Cas9 for the generation of the FUS R244C mutant line (pU6-gRNA-CMV-Cas9-GFP) was ordered from Sigma Aldrich. Both, the FUS R495X and TDP-43 M337V donor plasmids containing disease-linked and silent mutations were ordered by gene synthesis in pUC57 [General Biosystems] and encompass homology arms of 800 bp upstream and 800 bp downstream and 808 bp upstream and 815 bp downstream of the cleavage site, respectively. The donor plasmid containing the FUS R244C and silent mutations encompasses homology of 545 bp upstream and 505 bp downstream of the cleavage site, respectively. The pRR-EF1a-Puro FUS R495 plasmid was ordered by gene synthesis [General Biosystems]. To generate pRR-Puro TARDBP M337 two oligos containing the CRISPR/Cas9 target site (mdr789: 5'-CGC AGC ACT ACA GAG CAG TTG GGG GAC GT-3' and mdr790: 5'-CCC CCA ACT GCT CTG TAG TGC TGC GAG CT-3') were phosphorylated using T4 PNK, annealed by heating to 95 °C and cooling to room temperature creating 5' overhangs on both ends which were used to ligate the annealed oligos into the SacI, AatII sites of pRR-Puro[34,108]. pRR-Puro FUS R244C was generated analogously by annealing two oligos containing the CRISPR/Cas9 target site (5'-TGG TGG TTA CAA CCG CAG CAG TGG TGG CTA TGA ACC AGA GGT CGT GGA GGA CG T-3' and 5'-CCT CCA CGA CCT CTG GGT TCA TAG CCA CCA CTG CTG CGG TTG TAA CCA CCA GC T-3'). All plasmid sequences were confirmed by Sanger Sequencing.

## Human induced pluripotent stem cell (iPSC) culture

Ethical approval for the use of human iPSCs was obtained from the regional ethical review board in Stockholm, Sweden (Regionala Etikprövningsnämnden, Stockholm, EPN), and all work was conducted in accordance with local regulations. We obtained commercial DF6-9-9T.B[109] (Wicell) and KOLF2.1J (Jackson Laboratory, JIPSC001000) control iPSC lines[110]. Human iPSC lines were maintained with daily media changes in mTeSR1 or mTeSR-Plus medium (Stem cell Technologies, 85870 or 05825) as adherent cultures on plates coated with Matrigel (Corning, 354277) in a humid 5% CO$_2$ atmosphere at 37 °C. For passaging, they were treated with 10 μM Y-27632 (Tocris, 1254) for 1 h and detached in DPBS (ThermoFisher, 14190250) with 0.5 mM EDTA (ThermoFisher, 15575020) or ReLeSR (Stemcell Technologies, 05872). The Y-27632 treatment was maintained until the next media change.

## Genome editing

The DF6-9-9T.B control iPSC[109] (Wicell) was used to introduce all mutations using CRISPR/Cas9. The generation of FUS KO and FUS P525L iPSC lines has been described before[12,34] and genome editing was essentially performed as described therein. In short, to introduce the R495X mutation in exon 14 of the *FUS* gene two pCRISPR-EF1a-

eSpCas9(1.1)-R495 plasmids coding for the sgRNAs targeting the sequences 5'-GGG ACC GTG GAG GCT TCC GAG GG-3' and 5'-GGA CCG TGG AGG CTT CCG AGG GG-3' were used. To introduce the M337V mutation in exon 6 of the *TARDBP* gene a pCRISPR-EF1a-eSpCas9(1.1)-M337 plasmid coding for the sgRNA targeting the sequence 5'-GCA GCA CTA CAG AGC AGT TGG GG-3' was used. To introduce the FUS R244C mutation in exon 6, a pU6-gRNA-CMV-Cas9-GFP targeting the sequence: 5-ATG AAC CCA GAG GTC GTG GAG G-3' was used. One day before transfection, 10 μM Y-27632 [Stemcell Technologies] and 2 μM Pyrintegrin [Stemcell Technologies] were added to the stem cell media. On the day of transfection, 6 wells of a 6-well plate with 90% confluent parental control hiPSC in mTeSR1 containing Y-27632 and Pyrintegrin were transfected with TransIT-LT1 Transfection Reagent [Mirus] for the FUS R495X and Lipofectamine 3000 for TDP-43 M337V [Life Technologies] according to the manufacturer's instructions. For the FUS R495X editing, each well was transfected with a total amount of 4.67 μg of DNA, transfecting 190 ng of pRR-EF1a-Puro-R495 and 4480 ng of a mix of pCRISPR-EF1a-SpCas9-R495 (both targets mixed 1:1) and the R495X donor plasmid for HDR. For each well a different molar ratio of pCRISPREF1a-SpCas9 R945 and donor plasmid was used (1:1, 1:3, 1:6, 4:1, 3:1, 2:1). For the FUS R244C editing, three wells of iPSCs were transfected with 2ug of pU6-gRNA-CMV-Cas9-GFP, 4 ug FUS R244C HDR donor, and 1ug of pRR-Puro FUS R244C using Transfex [ATCC].

24 h post transfection, the medium was changed to mTeSR1 supplemented with the 5 μl of the HDR-enhancer L755507 [Sigma], 10 μM Y-27632, and 2 μM Pyrintegrin. 48 h post transfection, a single cell suspension was generated using Accutase [Thermo Fisher] and cells from the individual wells were pooled on a 15-cm plate in mTeSR1 containing 10 μM Y-27632 and 2 μM Pyrintegrin supplemented with 0.5 μg/ml Puromycin. Selection was maintained for one more day and Y-27632 and Pyrintegrin were maintained for four more days. For the R244C editing Pyrintegrin was omitted and selection performed with 0.25 μg/ml Puromycin in 10-cm plates. Thereafter, colonies growing from single cells were picked and gDNA was isolated for clone screening using TRIzol [Thermo Fisher] according to the manufacturer's instructions. The R495, R244C and M337 genomic loci were amplified from the genomic DNA using the KAPA Taq ReadyMix PCR Kit according to the manufacturer's instructions using primers sre10: 5'-GTG GCT CTC ACA TGG GTA AG-3', and sre11: 5'-AAA GAC CCA GAG TGG CTA AG-3' for the FUS R495 as well as FUS P525 locus, sre8: 5'-CAT GTT AGC CAG GAT GGT TTC G-3' and sre9: 5'-GCC AAT TCC TGG AAG CTG AAG TC-3' for the FUS R244 locus, and sre328: 5'-GAA TCA GGG TGG ATT TGG TAA TAG C-3' and sre331: 5'-AAT CCC ACC ATT CTA TAC C-3' for the TARDBP M337 locus. The PCR products were purified over a preparative agarose gel using the Wizard SV Gel and PCR Clean-Up System [Promega]. Purified PCR products were sequenced at Eurofins genomics or Source Biosciences with sre125: 5'-GGG TGA TCA GGA ATT GGA AGG-3' for FUS P525L, sre11: 5'-AAA GAC CCA GAG TGG CTA AG-3' for FUS R495X, mdr425: 5'-TTG TCC TTC ATT GCC TGG CAC TTG-3' for FUS R244C, and sre270: 5'-TGG TGT GTG GGA TGA ACT TTG-3' for TARDBP M337V, respectively. We also assessed the top five off-target sites for the editing of each cell line in the same fashion as the target sites (Supplementary Data 2). Genomic stability of the iPSC lines was assessed by screening with the hPSC Genetic Analysis Kit (StemCell Technologies; #07550) which did not indicate the presence of any genomic abnormalities in the top nine most reported genomic locations. Subsequent assessment with digital karyotyping (KaryoStat, ThermoFisher) identified a partial duplication of chromosome 14 in all cell lines, confirming comparable karyotypes across the isogenic lines.

## In vitro differentiation to heterogenous spinal neurons

Adherent human iPSCs were grown to near confluence before the differentiation to heterogenous spinal/hindbrain neurons was initiated by changing the media for two days to media d1-2 (200 nM LDN193189

[Tocris, 6053], 10 μM SB431542 [Tocris, 1614], 25 ng/mL SHH-C25II [R&D Systems, 464-SH-025], 250 nM Purmorphamine [Tocris, 4551], 1 μM retinoic acid [Sigma-Aldrich, R2625]), followed by seven days medium d3–9 (Medium d1-2 with 3 μM CHIR99021 [Tocris, 4423]), one day of medium d10 (Medium d3–9 with 200 μM ascorbic acid [Sigma-Aldrich, A4403], 125 μM db-cAMP [Tocris, 1141], 2.5 μM DAPT [Tocris, 2634], 10 ng/mL each of BDNF [R&D Systems, 248-BD-025], GDNF [R&D Systems, 212-GD-010], NT-3 [R&D Systems, 267-N3-025], and CNTF [R&D Systems, 257-NT-010]), and three days of medium d11–13 (3 μM CHIR99021, 25 ng/mL SHH-C25II, 250 nM Purmorphamine, 1 μM retinoic acid, with 200 μM ascorbic acid, 125 μM db-cAMP, 2.5 μM DAPT, 10 ng/mL each of BDNF, GDNF, NT-3, and CNTF). From day 14 onwards, the neuronal culture was maintained in medium d14+ (200 μM ascorbic acid, 125 μM db-cAMP, 2.5 μM DAPT, 10 ng/mL each of BDNF, GDNF, NT3, and CNTF) until the experimental time-points or harvest.

## In vitro differentiation to spinal motor neurons (homogenous protocol)

Specification of iPSC lines into spinal motor neurons was done as previously described[37], and was adapted from refs. 32,35 and detailed also here. Specifically, embryoid bodies were formed and neural induction initiated (days 1, 2) by resuspending iPSCs at 0.5–0.75 M cells/ml media in 50% v/v Neurobasal, 50% v/v DMEM/F12, 0.5% v/v N2 supplement, 1% v/v B-27 supplement, 5 μM Y-27632, 40 μM SB431542, 200 nM LDN193189, 3 μM CHIR99021, 200 μM ascorbic acid, followed by seven days (days 3–9) of ventralizing and caudalizing patterning in medium consisting of 50% v/v Neurobasal, 50% v/v DMEM/F12, 0.5% v/v N2 supplement, 1% v/v B-27 supplement, 200 μM ascorbic acid, 200 nM retinoic acid, 500 nM SAG [Tocris, 6390]. At day 10, the embryoid bodies were treated with 5 μM Y-27632 for 1 h, washed with DPBS [ThermoFisher, 14190250], and then dissociated for 5–10 min with gentle titration in warm TrypLE Express [ThermoFisher, 12604021], supplemented with 5 μM Y-27632. After the embryoid bodies disband into single cells, they were collected by centrifugation at 200 g for 4 min, and resuspended in maturation medium consisting of Neurobasal, B-27 supplement, 200 μM ascorbic acid, 10 μM DAPT, 10 ng/mL of each GDNF [Peprotech, 450-10] and BDNF [Peprotech, 450-02] and 5 μM Y-27632. The cell suspension was filtered through cell strainer caps and motor neuron progenitors were seeded at >15,000 cells/cm² on plates coated with fibronectin [ThermoFisher, 33016015], poly-ʟ-ornithine (Sigma-Aldrich, P4957), and laminin A (Sigma-Aldrich, L2020). The next day, the maturation media was replaced but Y-27632 was omitted. On day 12, remaining proliferative cells were eliminated with a 48 h pulse of 2 μM 5-fluorodeoxyuridine [Sigma-Aldrich, F0503]. From day 14 onwards, the neuron culture was maintained in medium d14+ consisting of Neurobasal, B27 supplement, 200 μM ascorbic acid, 10 ng/mL each of BDNF and GDNF with media changes every other day.

## Dissociation of neuronal cultures and single cell capture by FACS

Single cells from neuronal cultures were isolated as described[111]. Briefly, the adherent neuron cultures were treated with 5 μM Y-27632 (Tocris, 1254) for 1 h. After washing carefully with HBSS (ThermoFisher, 14025050), the cultures were dissociated in TrypLE Express (ThermoFisher, 12604021), supplemented with 5 μM Y-27632 for 20-30 min under gentle agitation and with pipetting at 5 min intervals to break cell clumps apart. The dissociated neurons were then collected by centrifugation at 200 g for 5 min and resuspended in 500 μL HBSS supplemented with 2% v/v B-27 supplement [ThermoFisher, 0080085SA], 2% v/v KnockOut Serum Replacement [ThermoFisher, 10828028], 2% w/v bovine serum albumin [Sigma-Aldrich, A9418-10G], 25 mM ᴅ-glucose [Sigma-Aldrich, G7021-100G], 5 μM Y-27632, 10 ng/mL BDNF [Peprotech, 450-02], 10 ng/mL GDNF [Peprotech, 450-10].

The cell suspension was then filtered through cell strainer caps into FACS tubes (BD Biosciences) and was kept on ice. The single cell suspensions were index sorted within an hour of harvest on a FACSAria III or FACSAria Fusion (BD Biosciences) at 4 °C and 20 psi through a 100 μm ceramic nozzle under sterile conditions. Based on TOPRO-3 iodide staining, dead cells were excluded and neurite fragments were removed by gates based on unstained live and stained dead-cell control samples. Single cells were collected into 96 multiwell plates (ThermoFisher, 11355960 and 10036339) by dispensing individual cells in 5 nL drops directly into 3 μL ice-cold single cell lysis buffer (scLB, 0.134% Triton X-100 [Sigma], 0.5 U/μL recombinant RNase inhibitor [Takara, 2313B], 3.33 mM each dNTP [ThermoFisher, R1121], 1 mM dithiothreitol [DTT, ThermoFisher], 1.67 μM Smarter-oligo-dTVN [Biomers.net], 6.7e-7 diluted ERCC Spike-In [ThermoFisher, 4456740]). After collection the plates were immediately sealed with adhesive sheets (ThermoFisher, AB0558) and snap frozen on dry ice.

## Single cell RNA sequencing (Smart-Seq2)

We prepared single-cell cDNA libraries using Smart-Seq2[109,112] with adaptations as described[111]. Cell lysates were thawed on ice before denaturation at 72 °C for 3 min. Following primer annealing for 2 min on ice, 2.76 μL master mix 1 (1 μL 5x Superscript II reaction buffer [ThermoFisher], 1 μL 5 M betaine [Sigma-Aldrich], 0.25 μL 100 mM dithiothreitol (DTT), 0.035 μL 1 M magnesium chloride [ThermoFisher], 0.1 μL 100 μM Smarter-TSO [Biomers.net], 50 U SuperScript II reverse transcriptase, 5 U recombinant RNase inhibitor) was added per lysate. On a thermocycler, the first strand synthesis was carried out as follows: 90 min at 42 °C; 10 cycles of 2 min at 50 °C, 2 min at 42 °C; and 15 min at 70 °C before cooling to 4 °C. Then 7.5 μL master mix 2 (1.15 μL) was added per sample. On a thermocycler, the second strand synthesis and amplification were carried out as follows: 3 min at 98 °C; 21 cycles of 20 sec at 98 °C, 15 sec at 67 °C, 6 min at 72 °C; and 5 min at 72 °C before cooling to 12 °C. The single-cell cDNA libraries were purified with SPRI beads (SeraMag beads, in 19.5% w/v PEG 8000, 1 M sodium chloride, 10 mM Tris-HCl pH 8.0, 1 mM disodium EDTA, 0.01% v/v Igepal CA630, 0.25% w/v sodium azide). The samples were barcoded with unique Illumina Nextera Indexes using reagents from the Nextera Sample Preparation Kit. The final libraries were then combined at equimolar concentrations before purification of the pools with SPRI beads. Sequencing was performed on HiSeq 2000 (Illumina) and NovaSeq 6000 (Illumina) at NGI Sweden, Stockholm.

## Single-cell RNA-Seq mapping and cell type classification

Raw sequencing reads were mapped to the hg38 human reference genome using STAR (version 2.7.0e)[113] with parameters "-out-SAMstrandField" set to "intronMotif". Aligned reads were assigned to genomic locations with the 'rpkmforgenes.py' script[114], using an annotation GTF file from Ensembl version 95 (GRCh38.95). In 'rpkmforgenes.py', the parameters "-fulltranscript" and "-rmnameoverlap" were used. After obtaining the raw counts, cells with <100,000 uniquely mapped reads and/or <5000 detected genes were excluded. Genes with an RPKM value of >1 were considered detected. Cells were classified as neurons based on expression of NEFM (neurofilament-medium) at >100 RPKM and SNAP25 at >10 RPKM. Neurons were further classified as motor neurons if they showed expression of at least one of the transcription factors *ISL1*, *ISL2* or *MNX1* at >1 RPKM, combined with either *SLC18A3* (VAChT) or *SLC5A7* (ChT) at >5 RPKM, but lacked expression of the interneuron markers VSX2 (< 1 RPKM) and SIM1(< 1 RPKM). All neuronal cells that were not classified as motor neurons were generically termed interneurons. V2a interneurons were neuronal cells that were positive for both *VSX2* (CHX10) and *SOX14* at >1 RPKM. *C9orf72*-ALS bulk RNA sequencing data[33] was obtained as fastq files and re-mapped to the human genome as described above.

## Identification of differentially expressed genes and pathways

Differential expression analysis between groups was conducted in R (version 3.6.1)[115] using the *DESeq2* package (version 1.24.0)[116]. Fit type was set to "local". An FDR-adjusted p-value of < 0.05 was considered significant. Gene set enrichment analysis was performed using the *fgsea* package version 1.8.0. DESeq2 outputs were ranked by test statistic to be used as input. These gene lists were compared to GO biological process pathways obtained from the human MsigDB collection (C5, GO-bp). Only pathways containing between 10-500 genes were considered in the analysis. UMAP plots, heatmaps and Venn diagrams were all generated in R. UMAP plots were generated using *Seurat* (version 4.2.0)[117] using either all cells or all neuronal cells. Batch normalization was conducted in Seurat using canonical correlation analysis (CCA). The dataset was separated in batches for each biological experiment, which were separately normalized using SCTransform and re-integrated using integration anchors and integration features. The top 2000 most variable gene features were used as integration features. Heatmaps were generated using the *pheatmap* package version 1.0.12. Venn diagrams were generated using the *Vennerable* package version 3.1.0 or the *venn* package version 1.11. UpSet plots were generated using packages *ggplot2* (version3.4.1) and *ggupset* (version 0.3.0). For QIAGEN's Ingenuity Pathway Analysis (IPA)[46] the list of differentially expressed genes between mutant motor neurons and control was used as input. Genes with FDR < 0.1 were considered significantly differentially expressed for the pathway analysis and were compared to the Ingenuity Knowledge Base reference set (version Q4 2022, accessed on February 7, 2023). Canonical pathways across the different sample sets were ranked according to the FDR value aggregate of pathway enrichment across the *FUS*-ALS MN datasets (internally calculated by IPA). Heatmaps of significance (FDR value) and activation scores (z-score) for the 12 highest ranking canonical pathways were generated. Pathways with an FDR of < 0.1 were considered significantly enriched. For mapping expression changes to protein structures, RCSB PDB entries for mammalian mitochondrial respiratory complexes I (5lc5, *Bos taurus*), II (1zoy, *Sus scrofa*), III (1bgy, *B. taurus*), IV (1occ, *B. taurus*), and IV (5ara, *B. taurus*) were retrieved, arranged and rendered in UCSF Chimera (version 1.16, build 42360) as in ref. [118]. The expression across human ALS motor neurons (mean $\log_2$ foldchange of ALS compared to control motor neurons) was used to colour the individual orthologous protein chains using a python script, generated in R (version 4.2.0) together with the scale legend.

## Immunofluorescence microscopy

For imaging of iPSCs, they were grown on 12-well chamber microscope slides [ibidi, 81201] and fixed with 4% paraformaldehyde for 20 min at room temperature. The cells were then washed three times for 5 min each with PBS and then permeabilized in TBS supplemented with 0.5% v/v Triton X-100 [AppliChem], 6% bovine serum albumin (BSA) [Sigma-Aldrich], in TBS for 30 min at room temperature. Primary antibody incubation was performed in antibody diluent (0.1% v/v Triton X-100, 6% w/v BSA, in TBS) overnight at 4 °C. The cells were washed three time for 5 min with antibody diluent before being incubated with secondary antibodies in antibody diluent for 2 h at room temperature. Subsequently, the cells were counterstained with 100 ng/ml DAPI in PBS for 10 minutes at room temperature and then washed twice for 5 min with PBS. Coverslips were then mounted using hard set antifade mounting medium [Vectashield, H-1400]. Images were acquired using an Eclipse Ti-2 epifluorescence microscope [Nikon] with a 20×/ dry lens.

Neurons were grown on glass coverslips or in imaging grade 96-well plates (ibidi, 89626; IBL Austria, 220.230.042) or in PhenoPlate 96-wells (revvity, 6055300) coated with fibronectin [ThermoFisher, 33016015], poly-L-ornithine (Sigma-Aldrich, P4957), and laminin A (Sigma-Aldrich, L2020) for immunofluorescence microscopy. At indicated time-points, the neurons were washed with warm HBSS

(containing magnesium and calcium chloride, ThermoFisher Scientific, 14025100), fixed in 4% w/v paraformaldehyde for 15 min, and washes three times with HBSS to end the fixation. Then, the neurons were permeabilized in blocking solution [0.5% w/v Triton X-100 (Sigma-Aldrich, X100-1L), 10% v/v normal donkey serum (Jackson ImmunoResearch, 017-000-121), in PBS (ThermoFisher Scientific, 70011044)] for 30 min. Primary antibodies were applied in fresh blocking solution overnight at 4 °C. Following three washes with blocking solution over 30 min, secondary antibodies diluted in blocking solution were applied for 1.5 h at room temperature in the dark. The neurons were washed twice with PBS (ThermoFisher Scientific, 70011044), followed by a single wash in ultrapure water (ThermoFisher Scientific, 10977049). The cells were mounted in Fluoromount G (ThermoFisher Scientific, 00-4958-02) and subjected to inverted confocal microscopy on an LSM700 (Zeiss) or LSM800-Airy microscope (Zeiss) or an Opera Phenix High Content Imaging system using the spinning disk confocal mode.

## Western blotting

One 10-cm dish of iPSCs was harvested by incubation with DPBS-EDTA and pelleted by centrifugation at 200 g. Cell pellets were resuspended in 400 µl RIPA buffer [Sigma Aldrich, R0278-50ML] supplemented with 2x Halt protease and phosphatase inhibitor cocktail [Thermo Fisher Scientific, 78441], 2 mM manganese sulfate, and 25 U/ml Cyanase nuclease [Serva, 18542.02] followed by incubation on ice for 30 min with vortexing in 5-min intervals. 400 µl of 2x LDS loading buffer (ThermoFisher Scientific, NP0007) supplemented with 250 mM DL-dithiothreitol (Sigma-Aldrich, 43816-50 ML) were added and samples were boiled 10 min at 70 °C. Samples were loaded on 4-12% Bolt Tris-Bis gels and ran with 1x MOPS buffer, followed by blotting on nitrocellulose membranes using the iBlot2 transfer device. Membranes were blocked for 1 h in TBS-Tween (0.1%) supplemented with 5% w/v non-fat dry milk, followed by incubation with primary antibodies overnight at 4 °C. After five subsequent washes with TBS-Tween (0.1%), secondary antibodies were added in TBS-Tween (0.1%) supplemented with 5% non-fat dry milk, incubated for 1.5 h at room temperature, followed by five consecutive 5-minute washes with TBS-Tween (0.1%) and scanning of the membranes on an Odyssey CLx scanner [Li-Cor].

## Quantitative PCR for mitochondrial copy numbers

Total DNA was isolated from motor neurons lyzed in Qiazol (Qiagen). The copy numbers of nuclear and mitochondrial genomes were determined by quantitative PCR using absolute quantification against PCR standards. The reactions consisted of 1x MESA GREEN qPCR MasterMix Plus for SYBR Assay (Eurogentec) with 3 ng total DNA and 0.5 µM of each primer in 15 µL volume and were run on an ABI 7500 Fast (Applied Biosystems). Primers for the nuclear genomic DNA (schwi637, 5′-TGC TTG CTC AGA AGG AGC TT-3′, and schwi638, 5′-TGG GTT CAG GAA CAG AGA CA-3′) amplify a region in Chr4p, whereas the primers for the mitochondrial genomic DNA (schwi818, 5′-CGC CAC CCT AGC AAT ATC AAC-3′, and schwi819 5′-AGG CTT GGA TTA AGG CGA CAG-3′, amplify a region in MT-CYTB. Analysis was performed in R (version 4.4.0).

## Extracellular flux XFe 96 Seahorse measurements

Human iPSC-derived motor neurons were differentiated according to the homogenous protocol and seeded at day 10 into Seahorse XF96-well plates (Seahorse Bioscience, Agilent Technologies). At day 21, intact and permeabilized cells were analysed. For intact cells, the plates were washed and incubated for 60 min in an air incubator at 37 °C with XF base medium (102353-100, Agilent), supplemented with 25 mM glucose (G6152, Sigma) and 2 mM glutamine (25030024, Gibco, ThermoFisher) (pH adjusted to 7.5). The XF96 plate was then transferred to a temperature-controlled (37 °C) Seahorse extracellular flux analyzer (Agilent Technologies) and allowed a temperature

equilibration period before assay cycles (consisting of 1-min mix, 2-min wait, and 3-min measure period). Oxygen consumption rates (OCR) and extracellular acidification rates (ECAR) were dissected into different functional modules as described in detail previously[119,120]. In brief, after four basal assay cycles oligomycin (5 μg/ml, O4876, Sigma) was injected to inhibit the ATP synthase to determine OCR related to ATP synthesis for 3-6 cycles (lowest value of 3-6 cycles was used for calculation). 2,4-Dinitrophenol (2,4-DNP; 100 μM, 34334, Sigma) was injected to stimulate maximal respiration, revealing maximal substrate oxidation capacity (3 cycles, highest value used for calculation). Next, rotenone (R, 4 μM, R8875 Sigma) and antimycin A (AA, 2 μM, A8674, Sigma) were added, followed by 3 assay cycles to determine the non-mitochondrial OCR (lowest of 3 values). Finally, 2-deoxy-glucose (2-DG, 100 mM, D8375, Sigma) was added to determine glycolytic extracellular acidification rates (ECARs), subtracting the lowest of 6 ECAR measurements after 2-DG injection. ECAR values were transformed to PPR values using the machine settings. PPR values after 2-DG injection were subtracted to correct for non-glycolytic acidification. After the measurement, the cells were lysed and total dsDNA amount per well was determined using Quant-iT PicoGreen dsDNA Assay Kit (P7589, Invitrogen, ThermoFisher). All OCR and PPR values were normalized to the DNA content of the individual well. The OCR/PPR values in the graphs were adjusted to the mean DNA content (dsDNA: 230 ng for DF-6-9-9T.B iPSC lines; 50 ng for KOLF2.1 J iPSC lines). For permeabilized cells, the plates were washed with mitochondrial assay solution (MAS) and incubated in a 37 °C air incubator in fresh MAS-buffer containing 0.006% Digitonin (D141,Sigma), 0.4% fatty acid-free BSA (A3803, Sigma) and 4 mM ADP (117105, Sigma) for 30 min, followed by the respirometric analysis. The following compounds (at final concentrations) were used: for Complex I: 10 mM pyruvate (P8574, Sigma), 1 mM malate (02288, Sigma); for Complex II: 10 mM succinate (S3674, Sigma), 2 μM rotenone; and for Complex IV: 10 mM ascorbate (A92902, Sigma) 100 μM TMPD (87890, Sigma), 2 μM antimycin A.

### Live cell imaging for tracking mitochondria
Motor neurons were derived using the homogenous protocol, but the embryoid bodies were attached at day 10 to imaging grade 24-well plates (ibidi, 82406) coated with fibronectin (ThermoFisher, 33016015), poly-L-ornithine (Sigma-Aldrich, P4957), and laminin A (Sigma-Aldrich, L2020) to promote axons outgrowth. The neurons were matured until at least day 27. For tracking mitochondria, motor neurons were cultured in Imaging Optimized BrainPhys medium (Stemcell Technologies, 05796) with SM1 NeuroCult supplement (Stemcell Technologies, 05711), 200 nM L-ascorbic acid (Sigma-Aldrich, A4403), and 10 ng/mL BDNF (Peprotech, 450-02) and 10 ng/mL GDNF (Peprotech, 450-10). After addition of 50 nM TMRM (ThermoFisher Scientific, I34361) mitochondria were labelled for 30 min before the medium was replaced. Live cell microscopy was performed in an Axio Observer 7 (Zeiss) equipped with a Plan Apo 40x/0.95 Ph3 objective and under a 5% $CO_2$ atmosphere at 37 °C. Four regions across distal axons were selected to record 5-min long time-lapses. For analysis of mitochondrial movement, individual mitochondria were traced in Fiji[121] using the TrackMate 7 extension with the StarDist detector[60] and a Sparse LAP tracker (as in 'mitotracker.py'). The mitochondrial traces were then annotated and analysed in R (version 4.2.0)[115] using our custom package 'mitotrackR' (https://github.com/schwi24/mitotrackR). Briefly, mitochondrial traces were split into branches to analyse mitochondria discernible over the time-lapse. Mitochondrial movement in the axonal direction was assigned using cone sections of a sphere with an internal angle of $2 \times \mathrm{acos}(1/3) \approx 141°$, that trisects the sphere into equal sections for classification of anterograde, retrograde, and orthogonal movement steps from the center. Stationary mitochondria were defined as discernible mitochondria that never exceed low migration speeds (maximal speed < 0.05 μm/s) or step in place so that they cover little distance over the time-lapse (summed up

distance < 2 μm). Discernible mitochondria with overall orthogonal movement were excluded from the analysis. For analysis of mitochondrial spacing across axons, the Euclidean distance of each mitochondrion to its nearest neighbouring mitochondria in the anterograde and retrograde direction was determined by the same cone section as described above. Statistical tests were performed in R (version 4.2.0).

### Meta-analysis of ALS-FUS case reports
We retrieved 115 articles or case reports from PubMed published between 2009 and 2022, from which the clinical features of 502 individual cases with ALS-FUS were collected and aggregated. We have used google translate and foreign language skills to extract information published in English, German, French, Chinese, and Japanese. If studies have reported sharing of identified cases, we merged them and indicated the multiple PMIDs. Unclarities and individual decisions during collection of the data from the original reports have been reported (Notes column) but were not systematically reviewed for this analysis. Notably, we have defined respiratory failure, the time of tracheostomy, or else death as the disease end point. All the data for the analysis is compiled in Supplemental Data 1. The analysis was conducted in R (version 4.2.0) and is available here, https://github.com/schwi24/Schweingruber_et_al_2023/tree/main/ALS_FUS_meta_analysis.

### Statistics & reproducibility
For quantitative assays in neuronal cultures, independent neuronal differentiation rounds were considered independent biological replicates and replicate wells considered technical replicates. Generally, no sample size estimations were performed and $n = 3$ was used unless stated differently in the figure legends or materials and methods. Wells in which neurons had detached were excluded from the analysis.

For single-cell RNA-Seq, neuronal cultures from a total of four independent differentiation rounds were used. *P*-values and statistics were derived from DESeq2 differential gene expression analysis, corrected for an FDR of 0.05. DESeq2 differential gene expression analysis was always performed as a two-sided test.

For live-cell imaging to assess mitochondrial motility in motor neurons, motor neurons from several individual differentiation rounds (namely control $n = 4$, FUS KO $n = 5$, FUS R244C $n = 3$, FUS R495X $n = 2$, FUS P525L heterozygous $n = 4$, FUS P525L homozygous $n = 5$, TARDBP M337V $n = 5$) were recorded in at least three attached spheroids per line.

### Reporting summary
Further information on research design is available in the Nature Portfolio Reporting Summary linked to this article.

## Data availability
We have deposited all raw and processed RNA sequencing data generated in this study on the NCBI Gene Expression Omnibus (GEO) under the accession number GSE226482, https://www.ncbi.nlm.nih.gov/geo/query/acc.cgi?acc=GSE226482. The *C9orf72*-ALS bulk RNA sequencing data were retrieved directly from the authors of the study[33]. Scans of fluorescent western blots, raw imaging files from confocal microscopy, the analysis files from Opera Phenix, qPCR data sets, and Seahorse assay result files with the following figshare link: https://su.figshare.com/projects/Early_mitochondrial_dysfunction_revealed_across_FUS-_and_TARDBP-ALS_at_single_cell_resolution/163252. The raw data has been deposited in figshare in the following hyperlinks. Figure 2a FUS localization in motor neurons: https://doi.org/10.17045/sthlmuni.22550041. Figure 2b, c Quantitative immunofluorescence microscopy for intracellular FUS localization in ALS motor neurons: https://doi.org/10.17045/sthlmuni.27952080. Figure 3d TDP-43 localization and phosphorylation in ALS motor neurons: https://doi.org/10.17045/sthlmuni.22551715. Figure 3e, f Quantitative immunofluorescence for TDP-43 localization in motor neuron lines:

https://doi.org/10.17045/sthlmuni.27939381. Figure 3j Cholinergic markers in the in-vitro derived motor neurons: https://doi.org/10.17045/sthlmuni.27940407. Figure 5h, l Mitochondrial protein expression by quantitative immunofluorescence microscopy: https://doi.org/10.17045/sthlmuni.27939792. Figure 5m, n Quantitative PCR to determine copy numbers of mitochondria in ALS motor neurons: https://doi.org/10.17045/sthlmuni.28152689. Figure 6 Overview of mitochondria in radially grown motor axons: https://doi.org/10.17045/sthlmuni.22553326. Figure 6 Tracking mitochondria in ALS motor axons: https://doi.org/10.17045/sthlmuni.22353466. Supplementary Fig. 2c Pluripotency markers in genome-edited iPSC lines: https://doi.org/10.17045/sthlmuni.22555465. Supplementary Fig. 2d FUS localization in genome-edited iPSC lines: https://doi.org/10.17045/sthlmuni.22555474. Supplementary Fig. 3b Motor neuron markers in heterogenous differentiation protocol: https://doi.org/10.17045/sthlmuni.22555564. Supplementary Fig. 3d Motor neurons markers in homogeneous differentiation protocol: https://doi.org/10.17045/sthlmuni.22555663. Supplementary Fig. 4c Interneuron marker expression: https://doi.org/10.17045/sthlmuni.27940434. Supplementary Fig. 5a FUS localization in ALS motor neurons (c-terminal antibody): https://doi.org/10.17045/sthlmuni.22555123. Supplementary Fig. 5b FUS localization in ALS motor neurons (P525 site-specific antibodies): https://doi.org/10.17045/sthlmuni.22555414. Supplementary Fig. 5c and 6b Western blots of FUS and TARDBP in iPSC lysate: https://doi.org/10.17045/sthlmuni.22555525. Supplementary Fig. 9d–g Metabolic profiling of ALS motor neurons: https://doi.org/10.17045/sthlmuni.22557340. Supplementary Fig. 9h Dynamic cultures of motor neurons: https://doi.org/10.17045/sthlmuni.27940497. Source data are provided with this paper.

## Code availability

All software for data analysis is publicly or commercially available and we deposited our custom code on github: https://github.com/schwi24/Schweingruber_et_al_2023/, and https://github.com/NijssenJ/Schweingruber2023. The meta-analysis of published clinical features of FUS-ALS cases is available here: https://github.com/schwi24/Schweingruber_et_al_2023/tree/main/ALS_FUS_meta_analysis. We published our scRNA-Seq analysis workflow and custom code here: https://github.com/NijssenJ/Schweingruber2023. The mitochondria tracking workflow and our custom code is available here: https://github.com/schwi24/Schweingruber_et_al_2023/tree/main/Mitotracker/, and the mitotrackR package, https://github.com/schwi24/mitotrackR.

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

## Acknowledgements

FACS was performed at the Biomedicum Flow Cytometry core facility at the Karolinska Institutet. We thank the operators Linda Pannagel and Juan Basil for their assistance. The authors acknowledge support from the National Genomics Infrastructure in Stockholm funded by Science for Life Laboratory, the Knut and Alice Wallenberg Foundation and the Swedish Research Council, and SNIC/Uppsala Multidisciplinary Center for Advanced Computational Science for assistance with massively parallel sequencing and access to the UPPMAX computational infrastructure. We thank the NGI project coordinators Mattias Ormestad and Elísabet Einarsdóttir. Confocal microscopy was performed at the Biomedicum Imaging Core (BIC) at the Karolinska Institutet and at the Imaging Facility at Stockholm University (IFSU). We thank the facility manager Chris Molenaar for his assistance. We thank the Wohl Cellular Imaging Centre at King's College London for help with quantitative microscopy with the Opera Phenix High Content Imaging system. We thank all our group members for helpful discussions throughout the project. This research was funded by the support from the Swedish Research Council (2020-01049), Radala Foundation, Ulla-Carin Lindquist's foundation for ALS research, Åhlen-stiftelsen (Åhlen's Foundation, grant numbers 233021, 223060, 213051, 203030, and 193042), the Olav Thon's Foundation, and the Swedish Brain Foundation (Hjärnfonden, grant number FO2021-0145) to E.H. C.S. was supported by an early post.doc mobility fellowship by the Swiss National Science Foundation (grant number 172233). J.A.B. was supported by a postdoctoral fellowship from the Swedish Society for Medical Research (SSMF). This research was also made possible through the support of the NOMIS Foundation, the UK Dementia Research Institute (award number UK DRI-6005 and UK DRI-6204) through UK DRI Ltd, principally funded by the UK Medical Research Council, Alzheimer's Society and Alzheimer's Research UK, the Medical Research Council (grant number MR/S025898/1), the John and Lucille van Geest foundation, and the Motor Neurone Disease Association (grant number 872-791) to M.-D.R. M.K. and M.J. are supported by the Novo Nordisk Foundation (grant number 0059646). Startup funding from the Department of Biochemistry and Biophysics at Stockholm University to E.H.

## Author contributions

E.H., M.-D.R.: conceptualization of study, supervision, and project administration. E.H., M.-D.R., M.J.: funding acquisition. E. H., M.-D.R.: supervision. E.H., M.-D.R., C.S., J.N., J.A.B., S.R.: methodology and study design. S.R., C.S.: literature review and meta-analysis of clinical features. S.R., J.M., N.O'.B., J.A.B., M.-D.R.: genome editing and characterization of iPSC lines. C.S., J.N., J.A.B.: experimental work on neuronal culture and single-cell sorting. C.S., J.A.B.: single cell library preparation. E.H., C.S., M.L., V.R.: quantitative immunofluorescence and analysis. C.S., M.K.: experimental work on metabolic assays (Seahorse). M.K.: metabolic analysis. C.S.: experimental work and analysis of mitochondrial motility. J.N., I.M.: RNA sequencing data processing, differential gene expression analysis, and gene set enrichment analysis. J.N., J.M., C.S.: Ingenuity pathway analysis. J.N., C.S., M.L.: preparation of figures. C.S., E. Hedlund: writing of original draft. All authors contributed to editing of the article and approved the manuscript.

## Funding

## Competing interests

The authors have no competing interests.
