## [Transparent Peer Review file · Nature Communications]

Single-cell RNA-sequencing reveals early mitochondrial dysfunction unique to motor neurons shared across FUS- and TARDBP-ALS

Corresponding Author: Professor Eva Hedlund

Version 0:

Reviewer comments:

Reviewer #1

(Remarks to the Author)

Schweingruber et al., utilize an elegant combination of iPSCs, gene editing, RNA sequencing and mitochondrial characterization (by Seahorse and motility assays) to nominate disruptions in mitochondria as a common feature across both TDP43 and FUS mutations that collectively contribute to up to 2-3% of all ALS cases. The study is well-designed and makes some intriguing observations that will be of interest to the field. It fails to provide any novel mechanistic insight (it's unclear how or why these mutations impact mitochondria respiration and motility) and there are several concerns that should be addressed to significantly increase the confidence in the results and the interpretations of the data that the authors make. Addressing these effectively would make the manuscript appropriate for publication at Nat Comms.

Major concerns

1. The CRISPR-edited iPSC lines should be QC-ed for off target (top 5 homology regions) and on-target genomic defects such as INDELS.
2. Only a very small proportion of MNs presented in Fig 1G appear to CHAT suggesting these are very immature or that the sc seq did not detect low levels of CHAT mRNA—the authors should use ICC to verify the identity of their cells.
3. What are the markers that clearly delineate “other” interneurons? Very few cells appear to express GAD2. What about GAD1? Again, performing some complementary ICC for GABA would be helpful in identifying the proportions of GABAergic neurons within these cultures.
4. Authors claim that transcriptional response is proportional to FUS mis-localization. No quantification is given for FUS mis-localization in the main figure 1, and it is not apparent by-eye if FUS R495X is more mis-localized than FUS P525L. Assessing the degree of N/C ratio in each case by ICC quantification will help.
5. Similarly, ICC quantification will help support the claims related to TDP43 in Fig 3: “The localization of TDP-43 did not vary between control and TDP-43 M337V motor neurons’ and ‘we observed a moderate increase in phosphorylated TDP-43 foci in the cytoplasm”.
6. Coupled Respiration is driven largely by metabolic demand. To decode respiratory chain intrinsic defects, they authors need to perform Seahorse Assays in permeabilized cells in a mitochondrial buffer. They should measure pyruvate + malate (complex I), succinate (complex II), TMPD+ ascorbate (Complex IV) driven respiration in the presence of DNP. This would assess whether parts of respiratory chain are significantly impaired.
7. Multiple studies have shown that impairment of mitochondrial function triggers integrated stress response (ISR). A key metabolic output of ISR is the one-carbon gene expression. Did the authors find activation of any ISR or one carbon metabolism enzymes? (See PMID: 27307216).
8. Fig 6 & Methods P29 line 2: Authors note that they normalize all OCR and PPR measurements to 230 ng dsDNA, which is

the mean dsDNA content across all wells measured. However, it is possible that in the 12 days between seeding and plating, there is differential survival of motor neurons between genotypes, and even within wells of the same genotype. Therefore, it is necessary to divide each OCR & PPR reading by the dsDNA content of each individual well to properly normalize for differences in density. For example, comparing the two main replicates of this seahorse assay, FUS P525L hetero neurons plated at the same density displayed substantially different absolute OCR.

9. Fig6c & fig6g & P12 line 31-32: Measurements of coupled ATP respiration depend on the metabolic energy demands of the cell, and are sufficient to prove deficits in complexes of the respiratory chain. Additionally, measurements of oxphos-driven ATP turnover rely on measures of coupled respiration. Authors claim 'ATP homeostasis is compromised due to defects in oxidative phosphorylation.' However, FUS GOF mutants may simply have reduced energy demands downstream of other dysfunctional aspects of cell biology (such as disrupted cytoskeletal transport; see fig7), rather than a bioenergetic respiratory chain failure being the cause of dysfunction. In order to make this claim, authors should assay the function of individual respiratory complexes by permeabilizing cells and recording OCR in buffer containing malate & aspartate (complex I), succinate (complex II), or N,N,N',N'-Tetramethyl-p-phenylenediamine dihydrochloride & ascorbic acid (complex IV).

10. Fig 6: Authors show summary calculations of mitochondrial respiration across multiple experiments. Close inspection of individual experiments reveals stark heterogeneity in the differences between control and mutated neurons. For example, when comparing parental control with FUS P525L homozygous neurons plated at the same density, one replicate shows nearly identical OCR traces, while the other shows some differences. It would be reassuring to have additional 3-4 replicates of these seahorse experiments to demonstrate that these differences are reliably reproducible. These should be coupled with QC on the neurons that are in the plates at the end of the experimental measurements i.e., WB or gene expression for MN markers.

Minor points

1. In Figs 1b and 3d clearly delineating the nuclear boundary would be helpful in assessing the degree of mislocalization.
2. P7 line 19-26: This long sentence is difficult to read without additional punctuation.
3. Fig 3b: 'Motoneuron' spelled as one word throughout panel.
4. Fig1i: Unclear to what comparisons the asterisks indicating significance level correspond. Are all the genotypes different from each other? Are the individual neurons types within each genotype different?
5. Fig2b/c: Authors reference the green portion of venn diagrams. To my eyes there is no green portion – the area appears yellow.
6. Fig3e/f legend descriptions for these panels are swapped.
7. Fig 7f & P14 line 1-5: I am unsure if Pearson's chi-square test is appropriate to compare these data. While the underlying data is categorical, it would seem the most appropriate to compare the overall percentage of stationary mitochondria per recording across genotypes (as is displayed in Fig. 7f) by ANOVA. The authors make the claim 'we found that FUS R244C, FUS R495X, FUS P525L (heterozygous and homozygous), as well as TDP-43 M337V motor neurons Schweingruber et al., 2023 14 displayed an increased number of stationary mitochondria compared to control motor neurons, unlike the FUS KO line.' However, inspection of figure 7f shows that the median percentage of stationary mitochondria of control and FUS R244C neurons are nearly identical. While it is appreciable that the spreads of these data are not equivalent, data displayed in 7f do not clearly enforce the differences noted in the text, whether the chi-square test is appropriate or not.

Reviewer #2

(Remarks to the Author)

The manuscript by Schweingruber et al identifies altered mitochondrial function and trafficking as a common disease process across multiple ALS mutations using iPSC-derived motor neurons. The study confirms what others have reported in mouse models, patients, and iPSC cultures in terms of mitochondrial malfunction which leaves important questions still unanswered. However, this paper does a nice job of looking across different FUS mutations (het, hom, KO) to untangle gain of function and loss of function properties as well as C9orf72 and TDP-43 to give a more robust look at ALS pathology. Finally, the inclusion of interneurons in some of the analyses was also unique and provides additional insight into cell-type specific processes. Overall, the paper was well done. Relatively minor comments remain.

1. Given that mitochondrial malfunction has been identified in multiple model systems including iPSCs, the impact of the results are somewhat blunted. As there are two approved ALS drugs that target mitochondrial function, does treating motor neurons with one of these drugs show benefit in the seahorse assay or for mitochondrial movement? What kind of phenotypes that can be observed in ALS iPSC-derived motor neurons (neurite outgrowth/retraction, cell death (maybe in response to oxidative stress), electrophysiological function, etc) can be restored by improving ATP levels or mitochondrial trafficking? These sorts of data could make the overall conclusions more significant for the field.

2. Although the transcriptional data don't indicate mitochondrial malfunction in interneurons, do FUS mutant interneurons show FUS mislocalization? Do they exhibit altered Seahorse results or mitochondrial trafficking defects? It appears that only motor neurons were examined for these phenotypes, so if interneurons are normal in these regards, then that would strengthen the cell-type specific phenotypes and support the conclusion "that cytoplasmic protein mislocalization gives rise to reduced mitochondrial activity in FUS-ALS and TARDBP-ALS." Otherwise, alternative interpretations would need to be explored.

Reviewer #3

(Remarks to the Author)

The authors addressed in this manuscript "Single cell RNA sequencing in isogenic FUS and TARDBP mutant ALS lines reveals early mitochondrial dysfunction as a common pathway in motor neurons" by Schweingruber et al. the mystery behind the cellular basis of FUS- and TARDBP-mutations in iPSCs using scRNA-seq, as well as a variety of experimental protocols, on genetically engineered iPSCs together with the genetically matched controls. The structure of the manuscript is easy to follow and the figures are aesthetically rather pleasing. The experimental work appears technically reasonable. The sample sizes used especially in the meta analysis are plausible. Though the amount of work and the research efforts put by the authors in this study are very much appreciated, there are serious technical and biological concerns that require further evaluation before publication of this work.

- Why was FACS used before scRNA-seq to enrich the neurons of interest? Transcriptome level changes occur rather fast in cells so any pre-treatment protocol, including FACS, before single cell sequencing would introduce a lot of stress to the cells, which would directly be reflected on the transcriptome, adding further bias to the final outcome. I understand this plate-based single cell approach requires a priori enrichment of cells but other single cell methods (10X, for instance) might likely to outperform this approach considering the particular hypothesis tested here.

- Also, mitochondrial dysfunction is already known to be one of the earliest pathophysiological events in ALS [doi:10.1016/j.neulet.2017.06.052]. It is not so easy to comprehend the novelty of this work or how exactly it would contribute to the current literature.

- Another generic issue would be how broad of a conclusion was drawn in this study. Instead of referring to some mitochondrial dysfunction phenotype in the primordial stages of ALS, more specific insights into, say, the mechanistic underpinnings of this phenomenon would be more fitting.

- Lastly, results from metabolic rate analysis using SeaHorse are known to be notoriously hard to reproduce on the bench-top as this is an extremely, maybe even too, sensitive of a technique. A complementary experimental approach to be included together with the SeaHorse would be more convincing for the reader. Also, I could not find anywhere in the manuscript the number of replicates used per sample group in this experiment. The results might indeed vary vastly as the sample sizes increase.

- Going further into details, the comparisons in Fig. 2B and 4B, -C, -E, and -F might include significance information of the intersection of individual pairwise comparisons to see the degree of overlap between any two gene lists by chance. This is because for some the "shared" number of DEGs identified after differential analysis is very low.

- The technical details of the meta-analysis mentioned in Extended Data 1 and S. Table 1 are completely missing in the Methods section. The technicalities of the protocol before and during the log-rank test should be mentioned in the manuscript because an important part of this study roots from that analysis.

- GSEA is based on a technically different algorithm from the other functional enrichment analyses available out there and it must be conducted taking the entire set of genes, not just the differentially expressed or the significant genes, as input (which is ranked by some feature in case of PreRankedGSEA). Using a much smaller input size significantly reduces the power of the prediction by this test. This misconception is unfortunately repeated many times over in the literature. Please consult the official GSEA page for further directions. It appears it is the DEGs that were ranked to perform this analysis, not all filtered genes. In this case, I doubt the statistical validity of all GSEA results as far as this study is concerned. I advise the authors to perform the same test taking all filtered genes as input for a given comparison and then see if the same gene sets are enriched in the input list or not.

- The number of remaining cells (i.e., 1415) after QC in the scRNA-seq pipeline is a rather low yield given the current standards these days. There is no discussion about it in the Discussion section as to why the authors ended up having these numbers after QC, nor any attempts to address this issue bioinformatically.

- IPA has a causal network analysis option to identify key targets as master regulators of the input genes. This is a rather reliable test one can leverage to gain new perspectives with regard to the mechanistic underpinnings of the mitochondrial dysfunction in ALS. It also helps in narrowing down the results for the next step of the downstream analysis. I would be curious what the results would have to say in this regard.

- A different representation of the GSEA results in Fig. 5B, -D, and -F would be appreciated as the current figure makes it really hard to understand the message that is being conveyed there.

- I particularly enjoyed reading about the mitochondrial motility assay and the results are biologically rather meaningful.

- I have no comments on the IF results: all looking reasonable and in line with the expected results.

Overall, there appears to be serious issues with the current version of the manuscript that needs to be addressed before publication. The language used throughout the manuscript is of high caliber. Yet, the novelty of the work is questionable even for the ALS-associated neurodegeneration specific to distinct mutations.

Version 1:

Reviewer comments:

Reviewer #1

(Remarks to the Author)

The authors have completed a thorough and responsive review to the prior concerns and their efforts should be acknowledged. Critically the manuscript has been significantly improved technically, with additional controls, cell lines, n's and refined analyses. At the same time however, the significance of the work has been weakened as the functional mitochondrial defects appear to be no longer apparent in the iPSC motor neurons. As things stand this work makes some interesting observations (some transcriptional convergence between FUS/TDP43/C9 mutations on mitochondrial genes, affecting MNs more so than interneurons) but the significance of these transcriptional alterations in mitochondrial pathways is unclear, given there are no functional impairments. Moreover, the mitochondrial motility alterations are typically related to defects in microtubule and trafficking regulatory pathways and thus the association between the transcriptional changes and reduced mitochondria motility remains unclear. Lastly, there appears to be no mechanistic insight into the connection between FUS and TDP-43 mutations and the phenotypes described in this work.

Reviewer #2

(Remarks to the Author)

The revised manuscript is very complete and well-supported. Prior concerns have been addressed.

Reviewer #3

(Remarks to the Author)

The authors have made significant improvements to the manuscript through thoughtful revisions. They have addressed the requested analyses wherever feasible and provided clarity on the novel aspects of their work. Specifically, they highlight how mitochondrial dysfunction in isogenic motor neurons serves as a contributor to gain-of-function mechanisms in FUS iPSC motor neurons. I have no further comments or requests.

Point-by-point response to the comments of the reviewers

We are very grateful to the reviewers for their insightful comments and suggestions which have greatly contributed to improving our manuscript. Below, we provide a point-by-point response in blue to their comments in black. Changes in the manuscript in response to comments are highlighted in blue.

Our original RNA sequencing data is submitted to GEO (Accession number GSE226482), reviewer access code: **crqfkcuqnlifhcj**

We have deposited our raw images and videos on FigShare please see the manuscript file for access links.

Reviewer #1 (Remarks to the Author)

Schweingruber et al., utilize an elegant combination of iPSCs, gene editing, RNA sequencing and mitochondrial characterization (by seahorse and motility assays) to nominate disruptions in mitochondria as a common feature across both TDP43 and FUS mutations that collectively contribute to up to 2-3% of all ALS cases. The study is well-designed and makes some intriguing observations that will be of interest to the field. It fails to provide any novel mechanistic insight (it's unclear how or why these mutations impact mitochondria respiration and motility) and there are several concerns that should be addressed to significantly increase the confidence in the results and the interpretations of the data that the authors make. Addressing these effectively would make the manuscript appropriate for publication at Nat Comms.

Major concerns

- The CRISPR-edited iPSC lines should be QC-ed for off target (top 5 homology regions) and on-target genomic defects such as INDELS.

Response: We thank the reviewer for this comment, and we fully agree. The results of the screening of the edited loci, were previously presented in the Extended Data figures and in previous publications that were referenced (Reber et al. 2018 EMBO J, PMID: 29167381; Jutzi et al. 2020. Nat Commun, PMID: 33311468). This data is now presented in **Extended Data Fig. 2a,b,e** and **Extended Data Fig. 6a**. We have now also conducted off-target analysis of the top 5 homology regions for each target as suggested. This analysis is now included in the updated manuscript as new **Supplementary Table 2**, and below as **Reviewer Table 1**:

Reviewer Table 1. Summary of gRNAs used for genome editing of iPSC lines, the top 5 homology regions and primers used for amplification and sequencing.

Mutation	Clone	Off-Target	Off-Target Sequence	Chromosomal location	Region	Gene	Forward Primer	Reverse Primer	WT	Cell line	Edited allele intact	Comment	
R495X.1	C13	1	GGGACGACGGAGGCTCCCTA	chr2:88832667	non-coding gene	LOC105374854	gctgcctgctcctcctcct	cttcctcctcactgagctc	Yes	Yes	YES		
		2	GGGACGACGGAGGCTCCGAG	chrX:150947272	intergenic	-	cctcccaacaccttgattttg	tgatgatgagaatttcagaga	Yes	Yes	YES		
		3	GGTACGATGAGGCTCCGAG	chr17:463318543	non-coding gene	LOC105371934	-	tcaggaaagttcagcctt	ttgagctcagctcctcc	Yes	Yes	YES	
		4	GGGACGATGAGGCTCCGAG	chr12:49706598	intergenic	-	-	tccaggtctccccaagac	aggaggatgtttggtat	Yes	Yes	YES	
		5	TGGACCGAGAGGCTCCGAG	chr3:65249932	intergenic	-	-	tcttctctctctctctctct	gtggatcaggtctctgta	Yes	No	NO	PCR of the locus did not result in amplicon (most likely deletion). Does not reside in an annotated gene
R495X.2	C13	1	GGGC-GAGGAGGCTCCGAG	chr2:231455249	protein-coding gene	NCL	cctaagggaagaggtctt	ccaacacgattgctcttg	Yes	Yes	YES		
		2	AGACATGAGGCTCCGAG	chrX:15742736	protein-coding gene	CASB	aggtttctcaagcttcca	gacggatcaccaccact	Yes	Yes	YES		
		3	GGACTGATGAGGCTCCGAG	chr1:112106724	intergenic	-	cttcagaaagttgagtc	acaagctcacacctccaaa	Yes	Yes	YES		
		4	GGGCTGGGACGCTCCGAG	chr21:37958111	non-coding gene	LOC105372798	-	gggctcaagttctctag	aaaggtgaccagcctacc	Yes	No	NO	Most likely deletion.
		5	GGA-CGCGGGGCTCCGAG	chr16:11569132	intergenic	-	-	tagaagacagaggtctcac	ggagactctaccctcaaa	Yes	Yes	YES	
P525L	C21 het C29 hom	1	GGAGCCAGGCTTAATA	chr8:42774230	intergenic	-	tgcttaagctcctccacagc	cttgaagccccaagatgt	Yes	Yes	YES		
		2	TAGAGCCAGTATAATA	chr8:16513651	intergenic	-	ttgttgaacctagctctcca	agaaaggaagcagaactact	N/A	N/A	N/A	Unable to amplify locus wildtype and edited cells	
		3	GCAAGCCAGTATAATA	chr5:169924449	protein-coding gene	SULT3	gacttttgcaactcagag	ttgtagggtggtggctg	Yes	Yes	YES		
		4	TGGATCCACTTAATA	chr4:13078433	intergenic	-	tcaggaaggaacagaaa	gaaagctaacgctctcttg	Yes	Yes	YES		
		5	GGATCCAGGCTTAATA	chr10:488469274	intergenic	-	ggaaacaactaaagctttga	gaagctctgcaaaacacctct	Yes	Yes	YES		
FUSKO	B4	1	GCGG-TGTCCACGAGCCT	chr21:4453274	protein-coding gene	TSPEAR	tggcacattcacagagaaa	cgcactgaggaaatctg	Yes	Yes	YES		
		2	CTGG-TGTCCACGAGCCT	chr2:128622810	intergenic	-	caatgagagagatgact	ttcaagagcagcagcagaa	Yes	Yes	YES		
		3	GTGAACCCACGAGCCT	chr11:114756882	intergenic	-	tgatgccaacacaaatga	agtgctcctcagaaactcag	Yes	Yes	YES		
		4	GTTGAGG-TCCACGAGCCT	chr2:230141054	intergenic	-	gattggccctctttctac	ctttggagccctctg	Yes	Yes	YES		
		5	GTC-ATGCTCCAGAGCCT	chr2:176992310	intergenic	-	aatcatctcttggctgct	ccatggcccaactctcca	Yes	No	NO	indel. Does not reside in an annotated gene	
R244C	R244C	1	GCTGTACCAGAGTCATGG	chr2:22631310	intergenic	-	ctccaggtctctctact	aggaacttttttggagga	Yes	Yes	YES		
		2	GA-AMACAGAGTCATGG	chr1:234898257	intergenic	-	tgttttaaagttccagctgc	acacacaggtgattgacc	Yes	Yes	YES		
		3	AATGAACCTAGAGTCATGG	chr1:66807950	intergenic	-	ggagtttagagatgagat	atgacagctttctcagct	Yes	Yes	YES		
		4	GAGGACCTAGAGTCATGG	chr10:4678071	intergenic	-	gctgtctgctcctatgc	agaaagctggaagatag	Yes	Yes	YES		
		5	GAGGA-GCAGAGTCATGG	chr4:40056667	protein-coding gene	N4BP2	agggcttgcctccagag	ggaactgcccgcctctcc	Yes	Yes	YES		
M337V	C25	1	CCAG-AATACAGCAGCTTG	chr22:26862932	intergenic	-	tg-aaggtggtctcccaca	aaactctgacgctcaact	Yes	Yes	YES		
		2	GCTGAC-ACAGCAGCTTG	chr22:261792426	intergenic	-	ggtctctctcttggagca	accacccaattctgca	Yes	Yes	YES		
		3	GCTGGACTCAGAGCAGTTG	chr8:31726684	intergenic	-	taagctccactggtgacct	gaaaggttggctctgaca	Yes	Yes	YES		
		4	TCAGATTACAGAGCAGTTG	chr16:8934267	intergenic	-	cactgcaataatcactctg	ccacgctggctctaaatc	Yes	Yes	YES		
		5	GCGAAGATAGAGCAGTTG	chr8:131809960	intergenic	-	tgcatcacacacacagcc	ctctggagcccaactg	Yes	Yes	YES		

The analysis showed that the FUS P525L lines, the FUS R244C and the TARDBP M337V lines are intact at the screened loci. In the FUS KO line we detected an indel (off target site 5). This indel resides in an intergenic region which we consider inconsequential.

In the FUS R495X line, which was generated by combining two gRNAs (R495X.1 and R45X.2) for editing, we identified two indels (off target site 5 for the 1st gRNA and site 4 for the 2nd gRNA). The off-target site 4 is not located in any coding region, but in a non-coding antisense RNA, KCN6-AS1 (or KCN6-IT1) and which would most likely result in a

deletion. We analyzed the expression of this antisense RNA in our RNA sequencing data and found that the expression is low (at the detection border). In fact, there is only one cell per group in the motor neuron, interneuron and V2a interneuron groups that expresses KCNJB_AS1. Apart from these three cells, it is not detected in our dataset, please see **Reviewer Figure 1**. We also analyzed the expression of the corresponding gene in the sense orientation, KCNJB6, which is stably expressed in motor neurons and remains unchanged in the FUS R495X line. Collectively, this shows that KCNJB6-AS1 is essentially not expressed in our cells and that this unintended genomic alteration does not affect our analysis.

We could not amplify and verify the loci for off-target 5 in the FUS R495X line 1 which indicates a genomic deletion. However, this off target site is located in an intergenic region which we consider inconsequential. In addition, our investigation on the effect of cytoplasmic FUS is performed across three cytoplasmic FUS lines. Thus, cell line-specific alterations caused by unintended genome editing artefacts are inherently mitigated.

Reviewer Figure 1. Analysis of KCNJB6_AS and KCNJB6 expression in iPSC-derived motor neurons, interneurons and V2a interneurons. RNA sequencing analysis shows that, (top panel) KCNJB6-AS is only expressed at low levels in one motor neuron, one interneuron and one V2A interneuron, while all other cells lack expression thereof. (bottom panel) KCNJB6 expression is expressed in motor neurons and not affected by the deletion in the KCNJB6_AS. (all plots are log transformed and thus the y-axis is a pseudo count, which means 0 expression).

2. Only a very small proportion of MNs presented in Fig 1G appear to CHAT suggesting these are very immature or that the sc seq did not detect low levels of CHAT mRNA—the authors should use ICC to verify the identity of their cells.

Response: We thank the reviewer for pointing out this matter and for giving us a chance to better demonstrate the presence of CHAT in our motor neurons. We classified MNs

in the single cell RNA sequencing data on the basis of their expression of the transcription factors ISL1 or ISL2 or MNX1 (at a cut-off > 1rpkm for each) as well as SLC18A3 (VACHT) or SLC5A7 (CHT), which is a combination of transcription factors and transporters that uniquely identify postmitotic MNs and has a strong basis in literature (**Extended Data Figure 3e**). Many of the MNs generated from the iPSCs express CHAT, but due to the color scale used in the heat map of **Figure 1g**, this is not so easy to see. We have now changed the color scheme of both **Figures 1g** and **Figure 1h** to make the expression patterns clearer, please see also below in **Reviewer Figure 2**.

Reviewer Figure 2. Gene expression of motor neuron and interneuron markers.

Furthermore, we are now also including violin plots demonstrating the expression of CHAT and VACHT in all single MNs and INs in extended Data Figure 3, as **New Extended Data Figure 3h**, (and below in **Reviewer Figure 3**).

Reviewer Figure 3. Violin plots of CHAT and VACHT expression in single MNs and INs.

As can be clearly seen in the updated heat map (**updated Figure 1g**, and **Reviewer Figure 2-3**) VACHT expression is always higher than CHAT. Interestingly, the VACHT gene is located within the first intron of the CHAT gene, in the same transcriptional orientation. In all our sequencing data, generated prior to this study, it is evident that the majority of reads from VACHT and CHAT align with the VACHT part of the gene, and do not

necessarily reflect expression levels. Please see included image from Allodi et al.2019 Stem Cell Reports, PMID: 31080111, where we sequenced MNs isolated from human post mortem tissues – here it is evident that CHAT is very low compared to SLC18A3 (VACHT) (**Reviewer Figure 4**). We would not argue against that our MNs are young as they were just generated a few weeks ago when we analyze them, but the pattern of gene expression is still highly similar to mature MNs in tissues and VACHT and CHAT are present.

Reviewer Figure 4. RNA sequencing of MNs isolated from Onuf’s nucleus, oculomotor nucleus (OMN) and spinal MNs from control patient tissues (each sample is a collection of around 50 MNs isolated individually and subsequently pooled and sequenced). Image is taken from Allodi et al, Stem Cell Reports, 2019, Supplemental Figure 3M.

To further solidify that CHAT is expressed in the MNs we generate in culture we have conducted deeper RNA sequencing of MNs from our ‘pure protocol’ in bulk and analyzed the reads covering the entire CHAT/VACHT locus, please see **New Extended Data Figure 3i**, and **Reviewer Figure 5** below.

Reviewer Figure 5. Bulk RNA sequencing and analysis of gene coverage of VACHT and CHAT in iPSC-derived MNs.

As suggested by the reviewer, we have also included immunofluorescent staining against CHAT protein in our cultures, which is shown in **New Extended Data Figure 3j** and in **Reviewer Figure 6** below.

Reviewer Figure 6. Immunofluorescent analysis of iPSC-derived motor neurons shows the presence of CHAT protein in ISL+ motor neurons.

3. What are the markers that clearly delineate “other” interneurons? Very few cells appear to express GAD2. What about GAD1? Again, performing some complementary ICC for GABA would be helpful in identifying the proportions of GABAergic neurons within these cultures.

Response: We appreciate this comment by the reviewer which has given us a chance to further clarify and delineate the interneuron subpopulations generated in our culture systems and why we grouped all together but the V2a interneurons. The “other interneurons” is a collective group in which we assemble diverse interneurons with low representation in the individual cell lines (at approximately 30 or fewer cells per line and group). Due to the low cell numbers were not confident we could use them individually for comparative analysis between the cell lines and have instead selected the well-represented V2a interneurons throughout the manuscript for this purpose. We have now conducted further analysis of gene expression within the V2a interneuron group and the broader “other interneuron” group. A portion of these “other interneurons” express high levels of ASCL1, but lack the V2a interneuron maker VSX2 (CHOX10), which indicates that these are V2b interneurons. Furthermore, there are also two distinct clusters that both express the V3 interneuron marker SIM1, but only one of them also expresses GAD1, **please see Reviewer Figure 7** and **New Extended Data Figure 4a,c** of the revised manuscript. From neurotransmitter synthesis, transporter and receptor analysis, it is clear that we have both glutamatergic and GABAergic interneurons in the

culture, see Reviewer Figure 7 and New Extended Data Figure 4b of the revised manuscript.

Selected genes (clustered and unclustered)

Neurotransmitter synthesis / transporters

Neurotransmitter receptors

Reviewer Figure 7. Interneuron marker expression and neurotransmitter transporters and receptors

4. Authors claim that transcriptional response is proportional to FUS mis-localization. No quantification is given for FUS mis-localization in the main figure 1, and it is not apparent by-eye if FUS R495X is more mis-localized than FUS P525L. Assessing the degree of N/C ratio in each case by ICC quantification will help.

Response: We thank the reviewer for this suggestion, and we have now systematically quantified FUS mislocalization across mutant FUS lines using an Opera Phenix high-content image screening system. This analysis, included in the revised manuscript as **Figure 2b-c (Reviewer Figure 8)**, demonstrates that both FUS R495X and FUS P525L homozygous motor neurons show a clear shift of FUS protein localization from the nucleus to the cytoplasm, while FUS R244C shows no shift as expected. Although there is no difference between the FUS R495X and the FUS P525L homozygous motor neurons, the FUS P525L heterozygous motor neurons have a lesser degree of mis-localization than the two other lines in line with previous reports (Kerk, et al. 2022. Stem Cell Rep, PMID: 35120624; Lenzi, et al. 2015. Dis Model Mech, PMID; 26035390).

Reviewer Figure 8. The intracellular FUS localization was quantified by high throughput immunofluorescence microscopy by measuring the compartmentalized FUS levels in ISL1⁺ motor neurons in 3D image stacks, followed by calculation of the nucleocytoplasmic ratio of FUS levels.

5. Similarly, ICC quantification will help support the claims related to TDP43 in Fig 3: “The localization of TDP-43 did not vary between control and TDP-43 M337V motor neurons’ and ‘we observed a moderate increase in phosphorylated TDP-43 foci in the cytoplasm”.

Response: We thank the reviewer for the suggestion and we have now added systematic quantification TDP-43 nucleocytoplasmic ratio as well as the level of cytoplasmic phospho-TDP-43 in TDP-43 M337V motor neurons compared to the isogenic control. This analysis which has been included in **Figure 3e,f**, of the revised manuscript (and **Reviewer Figure 9**) shows that the mutation did not cause a shift in nucleocytoplasmic ratio or increase the level of phospho-TDP-43 at the time that RNA sequencing analysis was done. This means that the changes we identified on a transcriptional level appears independent of protein mislocalization and aggregation and thus are an exceptionally early response to the mutation. We have now amended the text in the manuscript to reflect this matter, please see text in blue on pages 2, 3, 9, 10, 18 and 20 of the manuscript on this matter.

Reviewer Figure 9. Violin plots for the quantitative immunofluorescence measurements of intracellular TDP43 levels and phospho-TDP43 levels in ISL1⁺ TUBB3⁺ motor neurons. The nucleocytoplasmic ratio for TDP43 was calculated and shown in panel e, and the cytoplasmic levels of phospho-TDP43 is shown in panel f. The p-values were obtained from Wilcoxon rank sum test.

6. Coupled Respiration is driven largely by metabolic demand. To decode respiratory chain intrinsic defects, they authors need to perform Seahorse Assays in permeabilized cells in a mitochondrial buffer. They should measure pyruvate + malate (complex I), succinate (complex II), TMPD+ ascorbate (Complex IV) driven respiration in the presence of DNP. This would assess whether parts of respiratory chain are significantly impaired.

Response: We thank this reviewer for the insightful comment. Overall, previous significant differences in ATP-linked and maximal respiration were mitigated between the lines after increasing the n value with new experiments. With the new experiments, we also permeabilized the parental line, FUS R495X/ FUS P525L homozygous and TARDBP M337V and probed for differences in complex I (pyruvate/malate), complex II (succinate) and complex IV (ascorbate//TMPD) activity. Coherent with all newly analyzed measurements of the intact cells, no significant reduction of complex activity was seen compared to the parental line (**Reviewer Figure 10**). To further substantiate these conclusions, we performed similar measurements on commercially available KOLF2.1J lines (control, FUS R495X, and TARDBP M337V) on 8 independent plates, treating each plate as n=1 and correcting each well for DNA content. No significant differences were seen in respiration parameters of intact cells (neither in not normalized nor in DNA-normalized values). We also calculated the efficiency of mitochondrial energy transduction, termed coupling efficiency CE (Divakaruni and Jastroch 2022, PMID: 35971004), a parameter of mitochondrial functionality which is internally standardized, independent of variation of biological material, thereby reducing technical noise. None of the cell lines deviated significantly from the general efficiency of around 80%.

Reviewer Figure 10. Seahorse measurements of mitochondrial function on both intact and permeabilized iPSC lines across FUS and TARDBP mutations.

7. Multiple studies have shown that impairment of mitochondrial function triggers integrated stress response (ISR). A key metabolic output of ISR is the one-carbon gene expression. Did the authors find activation of any ISR or one carbon metabolism enzymes? (See PMID: 27307216).

Response: We thank the reviewer for pointing out these important pathways, which we have now investigated at the RNA level by GSEA between the control and mutant motor neurons. We did not find the one-carbon metabolism (OCM) or integrated stress response (ISR) significantly differentially expressed between control and our ALS mutant motor neurons at the transcriptional level. However, FUS KO motor neurons significantly downregulate the ISR pathway ($p_{adj} < 0.001$) (**Reviewer Figure 11**). This is an indication that the presence of normal FUS levels might already elevate ISR expression at a basal level.

We have further looked at the expression at the mRNA level of the four kinases that phosphorylate the translation initiation factor EF1 α and thus mediate the ISR, namely PERK, PKR, GCN2, and HRI. Among these four, both PERK and PKR are significantly upregulated at the mRNA level between control and FUS P525L homozygous motor neurons (**Reviewer Figure 12**). PERK integrates the ER-associated protein stress and PKR the cytoplasmic RNA stress into the ISR, which match the expectation of cytoplasmic protein and RNA aggregation in FUS-ALS. These conditions have been investigated following FUS overexpression, or stimulation of protein aggregation or stress induction in motor neurons (Szewczyk et al. 2023. Cell Rep. PMID: 36696267). However, in most of our homozygous FUS P525L motor neurons we have not observed apparent cytoplasmic FUS aggregation at the experimental time-points used. Thus, we conclude that the response is not yet triggered in our dataset, although there is evidence of the initial activation stages of ISR (at least for the FUS P525L homozygous motor neurons). If and how the changes we observe in our study are related to the ISR or OCM remains to be investigated but we could speculate that they may precede the onset of a fully activated ISR.

Reviewer Figure 11. GSEA of the pathways one carbon metabolism (OCM) and integrated stress response (ISR) between control and mutant motor neurons.

Reviewer Figure 12. Expression levels of EF1 α kinases that mediate the ISR.

8. Fig 6 & Methods P29 line 2: Authors note that they normalize all OCR and PPR measurements to 230 ng dsDNA, which is the mean dsDNA content across all wells measured. However, it is possible that in the 12 days between seeding and plating, there is differential survival of motor neurons between genotypes, and even within wells of the same genotype. Therefore, it is necessary to divide each OCR & PPR reading by the dsDNA content of each individual well to properly normalize for differences in density. For example, comparing the two main replicates of this Seahorse assay, FUS P525L hetero neurons plated at the same density displayed substantially different absolute OCR.

Response: We thank the reviewer for pointing out the flaw in our description. We normalized the OCR and PPR values individually per well and for further analysis, only used wells/plates, for which dsDNA content was assessed. All values (per ng dsDNA) were then multiplied with the mean dsDNA value of 230 ng, enabling the reader to judge the magnitude of OCR and PPR values in the figure. We have clarified this analysis in the material and methods (“All OCR and PPR values were normalized to the DNA content of the individual well. The OCR/PPR values in the graphs were adjusted to the mean DNA content (dsDNA: 230 ng for DF-6-9-9T.B iPSC lines; 50 ng for KOLF2.1J iPSC lines).”

9. Fig6c & fig6g & P12 line 31-32: Measurements of coupled ATP respiration depend on the metabolic energy demands of the cell, and are sufficient to prove deficits in complexes of the respiratory chain. Additionally, measurements of oxphos-driven ATP turnover rely on measures of coupled respiration. Authors claim ‘ATP homeostasis is compromised due to defects in oxidative phosphorylation.’ However, FUS GOF mutants may simply have reduced energy demands downstream of other dysfunctional aspects of cell biology (such as disrupted cytoskeletal transport; see fig7), rather than a bioenergetic respiratory chain failure being the cause of dysfunction. In order to make this claim, authors should assay the function of individual respiratory complexes by permeabilizing cells and recording OCR in buffer containing malate & aspartate (complex I), succinate (complex II), or N,N,N',N'-Tetramethyl-p-phenylenediamine dihydrochloride & ascorbic acid (complex IV).

Response: We have performed measurements on permeabilized cells as described in response to comment 6.

10. Fig 6: Authors show summary calculations of mitochondrial respiration across multiple experiments. Close inspection of individual experiments reveals stark heterogeneity in the differences between control and mutated neurons. For example, when comparing parental control with FUS P525L homozygous neurons plated at the same density, one replicate shows nearly identical OCR traces, while the other shows some differences. It would be reassuring to have additional 3-4 replicates of these seahorse experiments to demonstrate that these differences are reliably reproducible. These should be coupled with QC on the neurons that are in the plates at the end of the experimental measurements i.e., WB or gene expression for MN markers.

Response: We thank the reviewer for this comment and have added further replicates, as well as commercially available mutant cell lines. Altogether, the data suggest that the transcriptomic mitochondrial dysfunctions do not robustly translate into changes of oxygen consumption rates, please see **Reviewer Figure 10**. Potential compensatory effects may come with other functional pay-offs, such as changes in mitochondrial dynamics and motility.

Minor points

1. In Figs 1b and 3d clearly delineating the nuclear boundary would be helpful in assessing the degree of mislocalization.

Response: We have now added quantitative immunofluorescence analysis for the degree of mislocalization (**Fig. 2b,c**) as to point 4 above. We prefer not to obstruct the immunofluorescence image.

2. P7 line 19-26: This long sentence is difficult to read without additional punctuation.

Response: We agree and we have attempted to break up long sentences in the revised manuscript.

3. Fig 3b: 'Motorneuron' spelled as one word throughout panel.

Response: We thank the reviewer for spotting the misspelt labels. We have now corrected "motor neuron" in all the figures.

4. Fig1i: Unclear to what comparisons the asterisks indicating significance level correspond. Are all the genotypes different from each other? Are the individual neuron types within each genotype different?

Response: We have now added ticks to indicate the actual comparisons and updated the figure legends accordingly.

5. Fig2b/c: Authors reference the green portion of venn diagrams. To my eyes there is no green portion – the area appears yellow.

Response: We have renamed the color 'yellow' for the current **Fig. 2d,e**.

6. Fig3e/f legend descriptions for these panels are swapped.

Response: We thank the reviewer for spotting this mistake and have corrected it in the current manuscript.

7. Fig 7f & P14 line 1-5: I am unsure if Pearson's chi-square test is appropriate to compare these data. While the underlying data is categorical, it would seem the most appropriate to compare the overall percentage of stationary mitochondria per recording across genotypes (as is displayed in Fig. 7f) by ANOVA. The authors make the claim 'we found that FUS R244C, FUS R495X, FUS P525L (heterozygous and homozygous), as well as TDP-43 M337V motor neurons Schweingruber et al., 2023 14 displayed an increased number of stationary mitochondria compared to control motor neurons, unlike the FUS KO line.' However, inspection of figure 7f shows that the median percentage of stationary mitochondria of control and FUS R244C neurons are nearly identical. While it is appreciable that the spreads of these data are not equivalent, data displayed in 7f do not clearly enforce the differences noted in the text, whether the chi-square test is appropriate or not.

Response: We have assessed mitochondrial motility as proportions of stationary, anterogradely and retrogradely motile mitochondria. Such proportional data is inherently not normal, being heteroskedastic and bounded narrowly by zero and one (hundred percent). We think therefore that our statistical procedure is appropriate because it takes the linkage between the motility groups (anterograde, retrograde, and stationary) into consideration. ANOVA is generally not suitable for this data type and, more rigorously, and our data was not supported by tests of normality and homogeneity of variances (not shown in the manuscript, Shapiro-Wilk test $p = 0.00019$, Bartlett's test $p = 0.003223$). This can be appreciated in our new visualization of the mobility proportions as ternary plots (**Reviewer figure 13**), in which both the confinement and difference in variance between genotypes become obvious. In this new representation, it becomes evident that antero- and retrograde movement are generally balanced and thus line up on the anterograde-retrograde isoproportional line. Our main effect is in the proportion of stationary mitochondria (or motile mitochondria).

Reviewer Figure 13. Ternary plots with the proportions of mitochondria in mobility categories across the motor axons.

Reviewer #2 (Remarks to the Author)

The manuscript by Schweingruber et al identifies altered mitochondrial function and trafficking as a common disease process across multiple ALS mutations using iPSC-derived motor neurons. The study confirms what others have reported in mouse models, patients, and iPSC cultures in terms of mitochondrial malfunction which leaves important questions still unanswered. However, this paper does a nice job of looking across different FUS mutations (het, hom, KO) to untangle gain of function and loss of function properties as well as C9orf72 and TDP-43 to give a more robust look at ALS pathology. Finally, the inclusion of interneurons in some of the analyses was also unique and provides additional insight into cell-type specific processes. Overall, the paper was well done. Relatively minor comments remain.

Minor concerns

- 1 Given that mitochondrial malfunction has been identified in multiple model systems including iPSCs, the impact of the results are somewhat blunted. As there are two approved ALS drugs that target mitochondrial function, does treating motor neurons with one of these drugs show benefit in the Seahorse assay or for mitochondrial movement? What kind of phenotypes that can be observed in ALS iPSC-derived motor neurons (neurite outgrowth/retraction, cell death (maybe in response to oxidative stress), electrophysiological function, etc) can be restored by improving ATP levels or mitochondrial trafficking? These sorts of data could make the overall conclusions more significant for the field.

Response: First we want to thank the reviewer for the kind comments on our study. Regarding the 1st minor concern from the reviewer; Our new analysis clearly demonstrates a sequence of events in ALS pathology where mitochondrial motility is affected across all ALS mutations prior to any protein mislocalization, which has never before been described. It also highlights that this dysfunction is due to a gain of function mechanism for FUS. In this revised manuscript we highlight these novel findings. During the revision process we added a large number of Seahorse assay experiments which collectively showed the absence of significant effects on oxidative phosphorylation, which may be inherent to the irregularity of the iPSC-derived neuronal culture, (please see response to reviewer 2, on **page 9** of this response letter. We thus felt that any investigation of drugs that target mitochondrial function is outside of the scope of this study.

- 2 Although the transcriptional data don't indicate mitochondrial malfunction in interneurons, do FUS mutant interneurons show FUS mislocalization? Do they exhibit altered Seahorse results or mitochondrial trafficking defects? It appears that only motor neurons were examined for these phenotypes, so if interneurons are normal in these regards, then that would strengthen the cell-type specific phenotypes and support the conclusion "that cytoplasmic protein mislocalization gives rise to reduced mitochondrial activity in FUS-ALS and TARDBP-ALS." Otherwise, alternative interpretations would need to be explored.

Response: All cells exhibit FUS mislocalization in the FUS P525L homozygous line, for example, even iPSCs. We have amended our conclusion that mislocalization *per se* gives rise to reduced mitochondrial activity, even though the mutation and resulting mislocalization in motor neurons is correlated with increased transcriptional dysregulation compared to mutant FUS forms that do not cause mislocalization.

Reviewer #3 (Remarks to the Author):

The authors addressed in this manuscript “Single cell RNA sequencing in isogenic FUS and TARDBP mutant ALS lines reveals early mitochondrial dysfunction as a common pathway in motor neurons” by Schweingruber et al. the mystery behind the cellular basis of FUS- and TARDBP-mutations in iPSCs using scRNA-seq, as well as a variety of experimental protocols, on genetically engineered iPSCs together with the genetically matched controls. The structure of the manuscript is easy to follow and the figures are aesthetically rather pleasing. The experimental work appears technically reasonable. The sample sizes used especially in the meta analysis are plausible. Though the amount of work and the research efforts put by the authors in this study are very much appreciated, there are serious technical and biological concerns that require further evaluation before publication of this work.

Points

1. - Why was FACS used before scRNA-seq to enrich the neurons of interest? Transcriptome level changes occur rather fast in cells so any pre-treatment protocol, including FACS, before single cell sequencing would introduce a lot of stress to the cells, which would directly be reflected on the transcriptome, adding further bias to the final outcome. I understand this plate-based single cell approach requires a priori enrichment of cells but other single cell methods (10X, for instance) might likely to outperform this approach considering the particular hypothesis tested here.

Response: We use FACS to isolate individual cells, which are collected in individual wells for Smart-Seq2 that processes the single cell lysates as individual parallel reactions. We did not perform any selection or enrichment of neurons during FACS except for live cells and were strict on short processing times for our samples. As all cells were subjected to FACS any stress induced is similar across all cells and thus will not be enriched for. While other droplet-based sequencing methods (e.g. 10x) increase throughput and cell numbers drastically, this also comes at the cost of sequencing depth per single cell often limiting transcriptomic insight to few hundred genes per cell. We wanted to study even subtle early transcriptional dysregulation and opted for a well-established protocol with higher sequencing depth and thus detection of subtle differences also in lowly expressed genes.

2. Also, mitochondrial dysfunction is already known to be one of the earliest pathophysiological events in ALS [doi:10.1016/j.neulet.2017.06.052]. It is not so easy to comprehend the novelty of this work or how exactly it would contribute to the current literature.

Response: Although various forms of mitochondrial dysfunction have been described individually across genotypes (metabolic, morphological, etc), it is also worth noting that they are far from unified. Our systematic investigation in isogenic motor neurons revealed for the first time that various mitochondrial dysfunction is caused by gain-of-function mechanisms in FUS-ALS. We also show that this early dysregulation is unique to motor neurons and converges with TARDBP-ALS and C9orf72-ALS.

3. Another generic issue would be how broad of a conclusion was drawn in this study. Instead of referring to some mitochondrial dysfunction phenotype in the primordial stages of ALS, more specific insights into, say, the mechanistic underpinnings of this phenomenon would be more fitting.

Response: We thank the reviewer for pointing that we need to make the broader conclusion more clear. Based on additional experiments we have now modified both the abstract and text in results in discussion reflecting the broadly applicable finding that dysfunction in mitochondrial motility is shared across all ALS causations and uncoupled from protein mislocalization.

4. Lastly, results from metabolic rate analysis using SeaHorse are known to be notoriously hard to reproduce on the bench-top as this is an extremely, maybe even too, sensitive of a technique. A complementary experimental approach to be included together with the SeaHorse would be more convincing for the reader. Also, I could not find anywhere in the manuscript the number of replicates used per sample group in this experiment. The results might indeed vary vastly as the sample sizes increase.

Response: We fully agree with the reviewer that the sensitivity of the Seahorse assay requires further development. We have improved the analyses according to suggestions of reviewer 1. At this point, we show that the robust transcriptomic differences are not translating into changes of mitochondrial energy transduction. At this stage, we do not know if there is another pay-off to compensate the mitochondrial machinery to match the cell's ATP demand.

In detail, according to reviewer 1's valuable comment 8, we have increased the n value and removed all OCR data for which we could not assess DNA content. This substantially decreased differences of OCRs between the cell lines. Overall, significant differences in ATP-linked and maximal respiration were mitigated between the lines. With the new experiments, we also permeabilized the parental line, FUS R495X/ FUS P525L homozygous and TARDBP M337V and probed for differences in complex I (pyruvate/malate), complex II (succinate) and complex IV (ascorbate/TMPD) activity. Coherent with all newly analyzed measurements of the intact cells, no significant reduction of complex activity was seen compared to the parental line. To further correct for differences in biological material, we investigated the efficiency of mitochondrial energy transduction by calculating coupling efficiency CE (Divakaruni and Jastroch 2022, PMID: 35971004), a parameter of mitochondrial functionality which is internally standardized, independent of variation of biological material, reducing technical noise. None of the cell lines deviated significantly from the general efficiency of around 80%. To further substantiate our new conclusions, which differ from the first submission, we performed similar measurements on commercially available KOLF2.1J lines (control, FUS R495X, and TARDBP M337V) on 8 independent plates, treating each plate as n=1 and correcting each well for DNA content. No significant differences were seen in respiration parameters of intact cells (also in not DNA-normalized measurements). CE was remarkably stable between the three lines, around 80% efficiency.

5. Going further into details, the comparisons in Fig. 2B and 4B, -C, -E, and -F might include significance information of the intersection of individual pairwise comparisons to see the degree of overlap between any two gene lists by chance. This is because for some the "shared" number of DEGs identified after differential analysis is very low.

Response: We thank the reviewer for this suggestion and have performed pairwise set comparisons of DEGs as well as of the biological processes identified by GSEA. To this end, we have calculated the odds ratio between sets (of up- and downregulated genes

separately) to assess their overlap against the total set of all differentially expressed genes across cell lines. This data is presented in **Fig. 4g and Reviewer Figure 14**.

Others have previously used the Jaccard index for pairwise gene set comparisons (Ho et al. 2021 Cell Syst. PMID: 33382996). However, this measure can be dominated by differences in the set sizes. Indeed, our own DEG and GSEA sets vary noticeably in size as is often the case with transcriptomics data. We have therefore decided to use the odds ratio (OR) for the pairwise comparisons instead, a measure that puts the overlap sizes in relation to an expected value. Moreover, the odds ratio not only incorporates the idea of an expected value by chance overlap between sets of the given sizes (OR = 1) but also the ideas of attraction (OR > 1, enrichment) or repulsion (OR < 1, depletion) of elements into the overlap set against the total set (all differentially expressed genes across cell lines in our analysis). Thus, it translates intuitively to statistical testing in our opinion. We choose arbitrarily to highlight comparison pairs with at least 5-fold change in the OR and we have also calculated adjusted p-values using Fisher's exact test.

Reviewer Figure 14. Pairwise set comparison of dysregulated pathways across mutant motor neurons.

This analysis revealed overall that the overlap in up- as well as down-regulated pathways compared to the control across our mutant cell lines is larger than would be expected by chance selection of sets of the same sizes from the total set. This indicates greater similarity than expected by chance selection in our ALS lines. In contrast the overlap of the upregulated pathways between our lines and the C9 dataset from Mehta et al. 2021. Acta Neuropathol. PMID: 33398403, is lower than would be expected by chance selection, indicating more dissimilarity between these lines and ours. This might be due to genuine differences in dysregulation in C9orf72-ALS and FUS-ALS as well as TARDBP-ALS, due to different genetic backgrounds between our fully isogenic lines and their pairwise isogenic lines, due to deviation in the respective motor neuron differentiations, due to the mixed proportions of motor neurons in the bulk samples in contrast to motor neurons selected from our single cell dataset, or due to any combination of these.

Some individual comparison pairs are particularly interesting: The odds ratio between upregulated pathways in the FUS P525L heterozygous and FUS R495X motor neurons

(compared to the control line) is particularly high, indicating a greater degree of similarity in these sets than between others. Similarly, the odds ratio between FUS P525L homozygous and FUS KO line is much larger than between others, which is a further indication that a FUS LOF contributes more to the upregulated pathways in the FUS P525L homozygous line than in the other lines. This might support a notion of partial LOF in the FUS P525L homozygous line manifested in the upregulated pathways.

Although we think that these pairwise comparisons are revealing, we would also like to point out that the DEGs and differential pathways were of course selected by significance in differential expression analysis. So even if two sets in our analysis had not more overlap than expected by chance selection, they each still represent individual sets of significantly differentially expressed genes/pathways and may carry biological meaning. This should be kept in mind for the interpretation of these comparisons as well. Altogether, this analysis further justifies our approach of identifying shared dysfunction across lines in our opinion.

6. The technical details of the meta-analysis mentioned in Extended Data 1 and S. Table 1 are completely missing in the Methods section. The technicalities of the protocol before and during the log-rank test should be mentioned in the manuscript because an important part of this study roots from that analysis.

Response: We thank the reviewer for finding this unintended omission. We have added a full description to the methods section to correct that.

The case reports are not balanced overall and are biased due to the nature of single case reporting that tends to overrepresent more exceptional cases over mundane cases. Therefore, we have provided some basic feature characterization for the most reported FUS mutations in the Extended Data Figure 1. However, we did not attempt to correct for these biases in our analysis, e.g. by stratification according to sex or site of onset.

7. - GSEA is based on a technically different algorithm from the other functional enrichment analyses available out there and it must be conducted taking the entire set of genes, not just the differentially expressed or the significant genes, as input (which is ranked by some feature in case of PreRankedGSEA). Using a much smaller input size significantly reduces the power of the prediction by this test. This misconception is unfortunately repeated many times over in the literature. Please consult the official GSEA page for further directions. It appears it is the DEGs that were ranked to perform this analysis, not all filtered genes. In this case, I doubt the statistical validity of all GSEA results as far as this study is concerned. I advise the authors to perform the same test taking all filtered genes as input for a given comparison and then see if the same gene sets are enriched in the input list or not.

Response: We apologize for a potential misunderstanding in how the GSEA analysis was performed. We did not perform the analysis on solely the differentially expressed genes, as this is (as pointed out) not how GSEA is designed. Instead, we indeed used the entire set of expressed genes. We used the test-statistic of the DESeq2 output to rank the full set of tested genes (all protein-coding genes, $n = 15,960$), and subsequently tested the set of GO biological process terms (those including ≤ 500 genes) against this ranked list.

The *fgsea* package in R was used, using > '10/p-value threshold' # of permutations, and ultimately an adjusted p-value of < 0.05 was considered significant.

8. The number of remaining cells (i.e., 1415) after QC in the scRNA-seq pipeline is a rather low yield given the current standards these days. There is no discussion about it in the Discussion section as to why the authors ended up having these numbers after QC, nor any attempts to address this issue bioinformatically.

Response: We performed our experiments at the time when 96-well-based cell sorting and Smart-Seq2 sequencing was the state-of-the-art method, and before high throughput approaches such as 10x Genomics became a mainstream tool. In addition, the sensitivity of motor neurons after dissociation and FACS requires rapid freezing and processing of the sample, and does not allow for extended processing times. We believe our cell number provides an accurate representation of the motor neurons in culture, collected from $n > 3$ biological replicates, with large sequencing depth (compared to 10xGenomics) and high numbers of detected genes, which has enabled us to detect transcriptional dysregulation which would never have been identified through an approach with much more shallow sequencing depth, e.g. using 10x Genomics.

9. IPA has a causal network analysis option to identify key targets as master regulators of the input genes. This is a rather reliable test one can leverage to gain new perspectives with regard to the mechanistic underpinnings of the mitochondrial dysfunction in ALS. It also helps in narrowing down the results for the next step of the downstream analysis. I would be curious what the results would have to say in this regard.

Response: We thank the reviewer for this insightful suggestion, and we have now performed an upstream regulator analysis in IPA with the differentially expression data between control and mutant ALS motor neurons (**Reviewer Figure 15, Extended Data Figure 7**). Like the differentially expressed genes and pathways, in which we find first and mostly unique regulation, we identified mostly unique regulators in each individual dataset. However, we also identify more shared regulators between FUS P525L homozygous and TARDBP M337V motor neurons than the other lines and a large overlap between the ALS-FUS lines in general. To understand whether there are regulators that are shared across the ALS mutant motor neurons, which could underlie the shared dysfunction, we have focused on the protein-coding regulators in this intersection: CAB39L, DAP3, and LONP1 are all mitochondria-associated factors. Furthermore, we have calculated the activation z-score which indicate that CAB39L and DAP3 are significantly inhibited across our mutant motor neurons.

Out of curiosity about the regulation of potential dysfunction in the TARDBP M337V motor neurons, we have singled out the regulators in this line. Thus, we identified

TARDBP as significant upstream regulator, which is also significantly inhibited in this subset. This provides further evidence for a direct TARDBP dysfunction in the TARDBP M337V line.

Reviewer Figure 15. Upstream regulator analysis of the mutant motor neurons compared to control motor neurons. Upset plot of regulators across motor neuron lines. The colors indicated predicted activation or inhibition (or not determined).

10. A different representation of the GSEA results in Fig. 5B, -D, and -F would be appreciated as the current figure makes it really hard to understand the message that is being conveyed there.

Response: We have now represented the GSEA results in violin plots instead and hope that these better visualize the message, please see below in **reviewer Figure 16**.

Reviewer Figure 16. Updated GSEA results displayed as violin plots for Figure 5.

11. I particularly enjoyed reading about the mitochondrial motility assay and the results are biologically rather meaningful.

Response: We thank the reviewer for the positive response. We also included our analysis of mitochondrial distribution in motor axons to complement the motility data.

12. I have no comments on the IF results: all looking reasonable and in line with the expected results.

Response: We thank the reviewer for the positive assessment.

Overall, there appears to be serious issues with the current version of the manuscript that needs to be addressed before publication. The language used throughout the manuscript is of high caliber. Yet, the novelty of the work is questionable even for the ALS-associated neurodegeneration specific to distinct mutations.

Response: We hope that we have addressed the reviewer's concerns. We also hope that we have better highlighted the novelty of the work and the resource to the ALS community that our data encompasses.

REFERENCES

- Allodi et al. 2019 Stem Cell Reports, PMID: 31080111
Divakaruni and Jastroch 2022, PMID: 35971004
Ho et al. 2021 Cell Syst. PMID: 33382996
Jutzi et al. 2020. Nat Commun, PMID: 33311468
Kerk, et al. 2022. Stem Cell Rep, PMID: 35120624;
Lenzi, et al. 2015. Dis Model Mech, PMID; 26035390
Mehta et al. 2021. Acta Neuropathol. PMID: 33398403
Reber et al. 2018 EMBO J, PMID: 29167381;